# Microalgae Inoculation Significantly Shapes the Structure, Alters the Assembly Process, and Enhances the Stability of Bacterial Communities in Shrimp-Rearing Water

**DOI:** 10.3390/biology13010054

**Published:** 2024-01-19

**Authors:** Chen Lian, Jie Xiang, Huifeng Cai, Jiangdong Ke, Heng Ni, Jinyong Zhu, Zhongming Zheng, Kaihong Lu, Wen Yang

**Affiliations:** 1School of Marine Sciences, Ningbo University, No. 169 Qixingnan Road, Beilun District, Ningbo 315832, China; lianchen2021@163.com (C.L.); 19980215063@163.com (J.X.); 2211130089@nbu.edu.cn (J.K.); m17857816952@163.com (H.N.); zhujinyong@nbu.edu.cn (J.Z.); zhengzhongming@nbu.edu.cn (Z.Z.); lukaihong@nbu.edu.cn (K.L.); 2Fishery Technical Management Service Station of Yinzhou District, Ningbo 315100, China; huifengcai2023@163.com

**Keywords:** microalgae, bacterial lifestyles, bacterial communities, nutrient salts, shrimp-rearing water

## Abstract

**Simple Summary:**

It has been demonstrated that the adverse environmental issues induced by discharged water effluent from intensive shrimp farms could be alleviated through the inoculation of microalgae into shrimp-rearing water. However, the involvement of bacteria in this manipulated process and the role of microalgae in the bacterial community succession remain unclear. In a 30-day longitudinal study, two microalgae species (*Nannochloropsis oculata* and *Thalassiosira weissflogii*), in individual or in combination, were introduced into shrimp-rearing water to investigate the distinct effects of microalgal species and other environmental factors on bacterial community structure, assembly, and stability. Three key findings emerged from our study: (i) the introduction of different microalgae species shaped distinct structures in the bacterial community of the rearing water, with particle-attached bacteria responding more sensitively than free-living bacteria; (ii) the inoculation of both *N. oculata* or *T. weissflogii*, alone and in combination, could influence the contribution of stochastic processes in bacterial community assembly, potentially benefiting shrimp rearing; (iii) the addition of *T. weissflogii*, in particular, effectively enhanced the bacterial community stability, thereby establishing a healthier rearing environment. Those findings offer a new avenue for developing sustainable aquaculture practices.

**Abstract:**

Intensive shrimp farming may lead to adverse environmental consequences due to discharged water effluent. Inoculation of microalgae can moderate the adverse effect of shrimp-farming water. However, how bacterial communities with different lifestyles (free-living (FL) and particle-attached (PA)) respond to microalgal inoculation is unclear. In the present study, we investigated the effects of two microalgae (*Nannochloropsis oculata* and *Thalassiosira weissflogii*) alone or in combination in regulating microbial communities in shrimp-farmed water and their potential applications. PERMANOVA revealed significant differences among treatments in terms of time and lifestyle. Community diversity analysis showed that PA bacteria responded more sensitively to different microalgal treatments than FL bacteria. Redundancy analysis (RDA) indicated that the bacterial community was majorly influenced by environmental factors, compared to microalgal direct influence. Moreover, the neutral model analysis and the average variation degree (AVD) index indicated that the addition of microalgae affected the bacterial community structure and stability during the stochastic process, and the PA bacterial community was the most stable with the addition of *T. weissflogii*. Therefore, the present study revealed the effects of microalgae and nutrient salts on bacterial communities in shrimp aquaculture water by adding microalgae to control the process of community change. This study is important for understanding the microbial community assembly and interpreting complex interactions among zoo-, phyto-, and bacterioplankton in shrimp aquaculture ecosystems. Additionally, these findings may contribute to the sustainable development of shrimp aquaculture and ecosystem conservation.

## 1. Introduction

Aquaculture has experienced remarkable growth in the past few decades [1]. Among the rapidly expanding aquaculture industries, intensive shrimp farming has spread throughout many countries in Asia because of the high economic return of shrimp farming [2]. However, it is crucial to recognize that the most common rearing strategy used in shrimp farming— intensive shrimp rearing—can also lead to adverse environmental consequences, including the proliferation of animal or human pathogens, eutrophication of coastal ecosystems, and loss of biodiversity [3]. The dynamics of bacterial and microalgal communities, as well as their interactions, serve as both the culprits causing these environmental issues and the solutions for resolving them [4].

In shrimp-rearing ecosystems, bacteria, which are also present in various terrestrial and aquatic ecosystems, play a pivotal role in energy flows and nutrient cycling, ensuring a healthy and stable water microenvironment [5,6]. In addition, the bacterial community in rearing water also contains beneficial bacteria and pathogenic bacteria, which can affect the immune system of cultured shrimp, thus determining shrimp health and production [7]. Therefore, manipulating bacterial community dynamics is of utmost importance in urgently implementing relevant measures to mitigate potential environmental issues.

Among the array of improvement strategies, inoculation with microalgae stands out as a widely employed strategy for manipulating bacterial community structure in practical production [8]. Numerous studies have delved into the influence of microalgae on bacteria through both direct and indirect pathways, involving promoting or inhibiting bacterial growth through microalgal metabolites [9], and shaping the bacterial composition by altering the nutrient structure of the water [10]. Bacteria thriving in water exhibit two distinct lifestyles, namely free-living (FL) and particle-attached (PA) bacteria [11], each playing diverse and significant roles in bacterial community composition and function [12]. Although the influences of microalgae bloom on FL and PA bacteria in marine and freshwater ecosystems have been widely studied [13,14,15], the comparable and contrasting impacts of microalgae on FL and PA bacteria in aquaculture systems have been largely overlooked.

Bacteria with different lifestyles display notable differences in species composition and functional roles attributed to their unique habitats [16,17]. FL bacteria, existing in suspension in water and interact with dissolved organic particles, primarily contribute to the degradation of organic matter and the elimination of nitrogenous waste [18]. In contrast, PA bacteria engage in activities such as surface colonization, antipredation, and the improved acquisition of nutrients [17,19,20]. Consequently, understanding the successional mechanisms of these two bacterial communities and comprehending their interplay could provide novel insights into water quality management in shrimp aquaculture.

In addition to microalgae, nutrient factors constitute another critical influence on the bacterial community. Bacterial growth is heavily reliant on the nutrient structure, with distinct bacterial growth patterns being directed by specific nutrient factors [21]. For example, ammonia-oxidizing bacteria (AOB) obtain their energy by catabolizing un-ionized ammonia to nitrite. Nitrite-oxidizing bacteria (NOB) oxidize nitrite to nitrate [22]. The water quality environment is crucial for the formation of bacterial communities in rearing water [23]. Studies indicate that nitrogen, phosphorus, dissolved organic carbon, and their stoichiometric ratios can exert an influence on bacterial communities [24]. In addition, shifts in the bacterial community are also driven by changes in nutrients [25].

In the previous study [26], we found the unequal effects of microalgae inoculation and nutrients on bacterial communities in water during a 10-day experiment. To validate this phenomenon, we conducted a long-term experimental study. In the present study, we introduced two indigenous dominant microalgae (eustigmatophyte *Nannochloropsis oculata* and diatom *Thalassiosira weissflogii*), in individual or in combination, into shrimp rearing water. The aim of the present study was to reveal (1) the effects of different microalgae on rearing water quality; (2) the alterations in the structure of the rearing water bacterial community; and (3) the impacts on bacterial community assembly and stability. By establishing linkages among water nutrients, microalgae, and bacteria, we seek to identify the potential mechanisms of bacteria community mediated by microalgae. The findings from the present study are pivotal in safeguarding aquatic ecosystems and advancing environmental sustainability.

## 2. Materials and Methods

### 2.1. Microalgal Culture

The microalgae (*N. oculata* and *T. weissflogii*) were obtained from the Marine Algae Laboratory of Ningbo University, China. The microalgal culture solution was NMB3 used at a 1:1000 dilution. Autoclaved and cooled seawater was filtered through a 0.45 μm cellulose acetate membrane. The cooled seawater was used to prepare NMB3 medium for the cultivation of *N. oculata* [27]. For the cultivation of *T. weissflogii*, Na_2_SiO_3_ (2 mg/L) was added to NMB3 medium. The microalgae were mainly cultured in 5 L glass conical flasks with a light intensity of 100 mmol photons/(m^2^ s^−1^) at 27 °C. As the concentration of microalgae increased, the microalgae in the 5 L glass conical flasks were inoculated into a 10 L plastic cylindrical photoreactor for secondary culture of microalgae. The microalgae were used for subsequent experiments after reaching the exponential growth period, that is, when the *N. oculata* concentration reached 10^7^ cells/mL and the *T. weissflogii* concentration reached 10^6^ cells/mL.

### 2.2. Experimental Design

To simulate the practical shrimp-rearing environment, we obtained shrimp-rearing water from the middle and late stages of the shrimp ponds at Ningbo Xiangshan Lanshang Marine Technology Co., Ltd., Ningbo, Zhejiang Province, China (29°28′N, 118°6′ E), and transferred them to the production base of Ningbo University in Ningbo, China (29°46′ N, 121°57′ E) to be used in the experiments. After being transported to the production site, the shrimp-rearing water was left to settle for 10 days and then sequentially prefiltered through 100 μm and 1 μm sterilized nylon mesh to remove large particles and primary microalgae. Given the substantial difference in biological volume between *N. oculata* and *T. weissflogii*, microalgal biomass was opted as the indicator for experiment design and results presentation instead of abundance. Twelve 500 L polyethylene tanks were randomly divided into four groups: Group C (Control), Group N (addition of *N. oculate*, *N. oculata* concentration was 3 × 10^5^ cells/mL approximately), Group T (addition of *T. weissflogii*, *T. weissflogii* concentration was 8 × 10^3^ cells/mL approximately), and Group M (addition of *N. oculata* and *T. weissflogii*, where the concentration of *N. oculata* was about 1.5 × 10^5^ cells/mL and the concentration of *T. weissflogii* was about 4 × 10^3^ cells/mL). Each group had three replicates. Considering the difference in biological volume between *N. oculata* and *T. weissflogii*, based on the biomass of microalgae in shrimp aquaculture ponds, the initial biomass of microalgae was set at approximately 35 mg/L (the microalgal biomass of the shrimp ponds at Lanshang Marine Technology Co., Ltd.). In Group M, two microalgae with equal biomass were combined in tanks. The experiment was conducted under natural day and night conditions and the shrimp culture was conducted indoors in the transparent glasses (Ningbo University production site in Ningbo, China). All tanks were aerated using electric air pumps and daily water was exchanged throughout the entire experimental period. The experiment began on 25 April 2021, and was terminated on 23 May 2021.

### 2.3. Environmental and Bacterial Sample Collection

Water samples, each amounting to 500 mL, were collected from individual water tanks on days 0, 5, 8, 11, 15, 19, 22, and 29 of the experiment, totaling 192 samples (4 groups × 3 replicates × 8 time points × 2 duplicates). These samples were carefully preserved in sterile polyethylene bottles. All samples were stored in the dark at 4 °C and transported to the laboratory for processing. Within 3 h of sampling, each water sample underwent a two-step filtration process. Initially, it was filter-sterilized through a 3 µm pore size polycarbonate membrane (47 mm diameter, Millipore, Boston, MA, USA) to capture PA bacteria. Subsequently, the 3 µm filtrate underwent additional filter sterilization through a 0.22 µm pore size polycarbonate membrane (47 mm diameter, Millipore, Boston, MA, USA) to isolate FL bacteria. Two membranes of the same pore size from the same tank were placed in sterilization tubes and stored at −80 °C to serve as one bacterial sample. A total of 96 PA bacterial samples and an equivalent number of FL bacterial samples were gathered, resulting from the combination of 4 groups, 3 replicates, and 8 time points for each.

The water filtered through a 0.22 µm membrane was stored at 4 °C in 10 mL sterilization tubes. It underwent analysis using an automated spectrophotometer (Smart-Chem 450 Discrete Analyzer, Westco Scientific Instruments, Brookfield, WI, USA) to determine NH_4_^+^-N, NO_3_^−^-N, NO_2_^−^-N, and PO_4_^3−^-P within 48 h. Simultaneously, approximately 150 mL of water samples were transferred to a glass bottle and preserved with Lugol’s solution for storage. Microalgae were identified and quantified in sedimentation chambers (Hydro-Bios Apparatebau GmbH Kiel, Kiel, Germany) using an inverted microscope (CK2, Olympus Corporation, Tokyo, Japan), following the guidelines outlined in “Flora Algarum Marinarum Sinicarum” [28]. Phytoplankton biomass was computed through geometric approximations employing a computerized counting program known as OptiCount (https://science.do-mix.de/software_opticount.php, accessed on 1 September 2022).

### 2.4. Bacterial Illumina HiSeq Sequencing and Bioinformatic Analysis

DNA extraction from 192 bacterial samples (comprising 96 PA bacteria and 96 FL bacteria) was carried out using the MinkaGene Water DNA kit (Guangdong Magigene Biotechnology Co., Ltd., Guangzhou, China). During processing, a flocculant was added to adsorb impurities, and then centrifugation precipitated the flocculant and was improved to increase the purity of the nucleic acids. At the same time, the inhibitors were also adsorbed together. The concentration and purity of the extracted DNA were subsequently assessed using a NanoDrop One spectrophotometer (Thermo Fisher Scientific, Waltham, MA, USA). The V4 hypervariable region of the bacterial 16S rRNA gene underwent amplification utilizing Invitrogen-synthesized primers, namely 515F (5′-GTGCCAGCMGCCGCGGTAA-3′) and 806R (5′-GGACTACNNGGTATCTAAT-3′). Additionally, four equimolar PCR amplification products underwent purification and were then merged with the sequencing library, employing the NEBNext^®^ Ultra™ DNA Library Prep Kit designed for Illumina^®^ (New England Biolabs, Beverly, MA, USA). Ultimately, the libraries were subjected to sequencing on the Illumina HiSeq 2500 platform (Guangdong Magigene Biotechnology Co., Ltd., Guangzhou, China) to produce 250 bp paired-end reads.

The paired-end reads generated from sequencing were archived in the NCBI Sequence Read Archive under BioProject number PRJNA1029339 (https://www.ncbi.nlm.nih.gov/bioproject/PRJNA1029339, accessed on 7 December 2023). Bioinformatics processing of the sequencing data utilized USEARCH V.11 [29]. Initially, the paired-end reads underwent merging and denoising, employing the UNOISE3 algorithm to filter out short sequences that may result from sequencing errors or other artifacts [30]. The parameters were set to unoise_alpha = 2 and minsize = 4, following default settings. Subsequently, the filtered sequences were clustered into zero-radius operational taxonomic units (ZOTUs). ZOTUs of singletons, chimeric, mitochondria, and chloroplasts were removed. The low-quality reads, singletons, and chimeric sequences were decarded during sequence analysis process prior to clustered ZOTUs. Assignments of representative sequences to each ZOTU were performed using the RDP classifier, utilizing the SILVA bacterial database (version v138) with a 99% similarity threshold.

### 2.5. Statistical Analysis

Nutrient factors were analyzed using one-way ANOVA, and Tukey’s multiple comparisons were further employed for post hoc test in SPSS 27.0.1 if the ANOVA result was significant. Prior to the analysis, normality of the data were evaluated using the Shapiro–Wilk W-test, and homogeneity of variances was assessed. Graphs and plots were created using the ‘ggplot2’ package. For cluster analysis describing the similarity in bacterial composition between groups, the hclust() function in the “vegan” package was employed.

Principal component analysis (PCoA) was conducted using the Bray–Curtis distance to analyze the bacterial community. The analysis was executed using the cmdscale() function within the ‘ape’ package. Constrained principal component analysis (CPCoA) based on the Bray–Curtis dissimilarity metric was performed to visualize the overall structure of the bacterial community. This was achieved through the capscale() and cca() functions within the ‘vegan’ package. Permutational multivariate analysis of variance (PERMANOVA) was executed to identify community differences and assess influencing factors. The adonis() function within the ‘vegan’ package was employed for this analysis. To test the statistical significance of differences between bacterial communities, analysis of similarity (ANOSIM) was performed using the anosim() function in the same ‘vegan’ package. The multiresponse permutation procedure (MRPP) was executed using the mrpp() function in the ‘vegan’ package. To examine the relationships between bacterial communities and nutrient factors, redundancy analysis (RDA) was applied. The significance of environmental factors was assessed through a permutation test. Significantly correlated environmental factors were identified using the Spearman Mantel test. The neutral community model (NCM) gauges the contribution of stochastic processes to community assembly by predicting the relationship between the occurrence frequency of ZOTUs and their relative abundances. The R^2^ value assesses the overall fit of the neutral community model to the microbiota. Meanwhile, the Nm value represents the dispersal capacity of the entire microbiota, the N value indicates the size of the microbiota, and the m value signifies the species migration rate. Community stability was assessed using the average variation degree (AVD) index, calculated as the deviation degree from the mean of normally distributed OTU relative abundances among different groups. The variation degree for each OTU was calculated using the following equation (Equation: $\left|a_{i}\right|=\frac{\left|x_{i}-\bar{x_{i}}\right|}{\delta_{i}}$), in which $a_{i}$ is the variation degree for an OTU, $x_{i}$ is the rarefied abundance of the OTU in one sample, $\bar{x_{i}}$ is the average rarefied abundance of the OTU in one sample group, and $\delta_{i}$ is the standard deviation of the rarefied abundances of the OTU in one sample group. The AVD values were calculated using the following equation (Equation: $AVD=\frac{\sum_{i=1}^{n}\frac{\left|x_{i-\bar{x_{i}}}\right|}{\delta_{i}}}{k\times n}$), in which *k* is the number of samples in one sample group, *n* is the number of OTUs in each sample group [31].

## 3. Results

### 3.1. Variations in Microalgal and Nutrient Factors

The changes in nutrient salts and microalgal content differed among the treatment groups (Table 1 and Appendix A). Notably, differences were observed in all nutrient salts, except for NH_4_^+^-N and NO_2_^−^-N and their corresponding changes. In absolute values, the groups with microalgal inoculation exhibited higher nutrient levels than the control group. Furthermore, when considering the changes in nutrient salts, Groups N, T and M displayed higher levels than Group C. Regarding the impact of microalgal inoculation, the biomass of *T. weissflogii* decreased and *N. oculata* remained stable in the combined microalgal inoculation treatment.

### 3.2. Dynamics of Bacterial Community Composition and Diversity

#### 3.2.1. Alpha Diversity and Taxonomic Composition

A total of 192 samples were analyzed for α-diversity (Appendix A and Figure 1a). The results showed that similar α-diversity indices were observed within the same lifestyles. However, there were significant differences in the α-diversity indices among different lifestyles. The abundance of bacterial phyla or classes was derived for different lifestyles based on sampling time (Figure 1b). In terms of bacterial communities, *Actinobacteriota* increased and then decreased, and *Alphaproteobacteria* decreased and then increased with time. In terms of lifestyle, there were more *Verrucomicrobiota* and fewer *Patescibacteria* in PA than in FL. Additionally, with the inoculation of microalgae, *Campylobacteria* decreased (Figure 1b).

#### 3.2.2. Beta Diversity

At the outset of the experiment, our analysis using MRPP, ANOSIM and Adonis indicated a degree of similarity among the bacterial communities within the four groups (Appendix A). However, as the rearing progressed, PCoA based on Bray–Curtis dissimilarity of all samples revealed that both PA and FL bacterial communities varied over time (Figure 2a). Importantly, their temporal patterns differed significantly [ANOSIM, R^2^ = 0.598, *p* = 0.001]. PERMANOVA indicated that variations in the bacterial community were primarily influenced by rearing time, treatment, and lifestyle (Table 2). Therefore, the four groups had different temporal patterns [ANOSIM, R^2^ = 0.0512, *p* = 0.001]. Similarly, in the CPCoA of the PA and FL bacterial communities (Figure 2b–d), a clear impact of microalgal inoculation on the community structure was observed.

### 3.3. Differences in Species among Treatments

In the four treatment groups, the bacterial community formed distinct evolutionary branches, encompassing 6 phyla, 7 classes, 19 orders, 23 families, and 26 genera. The major planktonic bacteria identified were *Proteobacteria*, *Bacteroidetes*, *Pseudomonadota*, *Actinobacteriota*, *Firmicutes* and *Verrucomicrobia* (Figure 3a). The results of the LDA revealed that *Proteobacteria*, including *Aeromonadaceae* and *Oceanisphaera*, were notably abundant in the control group. In the group where *N. oculata* was added, *Bacteroidetes*, including *Marinagarivorans*, emerged as key microorganisms. Conversely, in the *T. weissflogii* group, *Pseudomonadota*, including *Rhizobiaceae*, was abundant (Figure 3b). Spearman’s correlation coefficients were calculated between representative bacteria and each environmental factor. Every genus was significantly correlated with at least one nutrient factor or microalgae (Figure 3b).

### 3.4. Relationship between Bacterial Community Structure and Environmental Factors

To unravel the intricate interplay between the bacterial community, nutrient factors, and microalgae, a redundancy analysis (RDA) was performed, encompassing all factors (including nutrient salts and their amount of change and microalgae) and the bacterial community (Figure 4). The results underscored the significant influence of NO_3_^−^-N, PO_4_^3−^-P, NH_4_^+^-N, NO_3_^−^-N_change, NO_2_^−^-N_change, PO_4_^3−^-P_change, and *N. oculata* on the water bacterial community. Among these impactful factors, environmental variables and their changes exerted a more substantial influence than *N. oculata*. In terms of lifestyle, PA bacteria exhibited a more pronounced response than FL bacteria (Table 3).

### 3.5. Assembly Process and Stability of Microbial Communities

The neutral model provided a robust description of the microbial taxa occurrence frequency within individual communities. Notably, stochastic processes accounted for 58%, 59%, 49.3%, and 53.8% of the variation in bacterial communities for the control, *N. oculata*, *T. weissflogii*, and combined microalgae groups, respectively. Particularly noteworthy was the significantly higher representation of the microalgae-added groups than the control group (Figure 5). The cumulative relative abundance of neutrally distributed ZOTUs in the control group surpassed that in the microalgae-added groups (C, 54.4%; N, 25.8%; T, 25.4%; M, 24.5%). This disparity indicates that the addition of microalgae impacts the stochastic processes governing the bacterial community.

A neutral model-based analysis at the family level was conducted in the upper (ZOTUs occurring more frequently than predicted), middle (ZOTUs considered neutrally distributed), and lower (ZOTUs occurring less frequently than predicted) sections (Appendix A). In our study, *Devosiaceae* and *Idiomarinaceae* exhibited higher relative abundances in the control group than in the microalgae-added groups. Conversely, *Rhodobacteraceae* showed a significantly higher abundance in the microalgae-added groups than in the control group. This pattern was consistent for *Flavobacteriaceae* and *Sphingomonadaceae*.

Bacterial community stability was assessed using the average variation degree (AVD) (Figure 6). Lower AVD values signify higher microbial community stability. Significant differences were observed among the different treatment groups. Regarding lifestyle, the AVD of PA bacteria was notably lower than that of FL bacteria. Remarkably, among PA bacteria, the group with *T. weissflogii* addition exhibited the lowest AVD.

## 4. Discussion

Several studies have highlighted the positive impact of inoculating beneficial microalgae on the microenvironment of shrimp-rearing water and the health of shrimp. However, the mechanisms underlying how different microalgal species affect aquatic environments and their interaction with bacterial communities of different lifestyles remain relatively unexplored. In this study, we investigated this by separately and collectively inoculating shrimp-rearing water with two dominant indigenous microalgal species. Our findings revealed that *N. oculata* exerted a more pronounced influence on nutrient levels than *T. weissflogii* and the combination of the two species. Moreover, the inoculation of microalgae induced alterations in the composition and assembly processes of bacterial communities. Notably, when *T. weissflogii* was inoculated, the bacterial communities displayed a higher degree of stability. Additionally, we also delved into the distinct responses of bacterial communities with diverse lifestyles to changes in microalgae and nutrient levels.

### 4.1. Microalgae Inoculation and Nutrient Changes Distinctly Influence the Response Patterns of PA and FL Bacterial Communities

Microalgae exert a strong influence on the diversity and composition of bacterial communities, as indicated in previous studies [32]. Our results corroborate those findings, showing that significant shifts in the rearing water bacterial community structure occurred with the inoculation of microalgae (Appendix A). Importantly, different changes in bacterial community structure occurred with the addition of different microalgae (Figure 3). The key bacteria in the control group were *Alphaproteobacteria*, *Actinobacteriota*, *Firmicutes,* and *Bacteroidetes*. However, with the addition of microalgae, a new key bacterial group, *Gammaproteobacteria*, emerged within the bacterial community (Figure 3). This finding aligns with previous observations where microalgae, specifically *N. oculata*, influenced bacterial community structure [27]. In addition, there were differences in the effects of different microalgal treatments on certain bacteria. Chlorophyta influence bacterial communities through metabolites [33]. A *N. oculata*–bacteria consortium with nonpathogenic r-strategy bacteria affiliated with the family *Rhodobacterales* could prevent opportunistic bacteria affiliated with the order *Alteromonadales* by competitive exclusion [34]. In addition, interactions have also been observed between diatoms and associated bacteria. In microalgae-inoculated waters, *T. weissflogii* could form a symbiotic relationship with specific bacteria [35]. Among all the factors with significant effects, our analysis indicated that nutrient factors had a stronger influence on the bacterial community than microalgae (Table 3). Previous studies have shown the dynamic nature of the water microbiota during shrimp culture, where environmental factors, especially nitrogen and phosphorus, play a pivotal role in these microbial fluctuations [21]. Nitrifying and denitrifying bacteria are mainly found in the phylum *Proteobacteria* [36]. Our results have shown that among the remarkable bacteria, *Roseovarius* are nitrifying bacteria [37] and *Paracoccus* are denitrifying bacteria (Figure 3) [38]. When the nitrogen source is ammonium, the oxygen generated by microalgae has a positive influence on nitrifying bacteria [39]. Conversely, when nitrate serves as the nitrogen source, oxygen can have an inhibitory effect on denitrifying bacteria [39]. Thus, on the one hand, microalgae directly affect bacterial communities through metabolites [40]. On the other hand, microalgae indirectly affect bacterial communities through nutrient factor mediation [41].

Bacterial communities with different lifestyles may play different roles in biogeochemical cycling [42]. Some previous studies have demonstrated that FL and PA bacteria are significantly different from each other [16,43]. In our study, the composition of PA and FL bacterial communities in shrimp-farming waters was significantly different (Appendix A). Microalgae and environmental factors directly affected the PA community (Figure 2). The PA bacterial community responded to microalgae and nutrients more than the FL bacterial community (Table 3). Therefore, we inferred that bacterial communities with different lifestyles have different response patterns to microalgae and nutrient factors. PA bacteria exhibit a higher richness in bacterial communities compared to FL bacteria, and this phenomenon may be due to the transfer of the FL bacterial community from the PA bacterial community [44]. PA bacteria occupy a higher ecological niche in their microbial environment [45], and likely possess greater stability and resilience to environmental change than their FL counterparts [46]. In summary, these research findings correspond to those of our study.

### 4.2. Microalgae Inoculation Induces Alterations in the Assembly Processes of the Bacterial Community and Enhances Community Stability

Several factors influence bacterial community structure, broadly categorized into two groups: deterministic factors and stochastic factors. Deterministic factors involve competition and niche-specific variables, while stochastic factors encompass microbial dispersal resulting from events such as colonization/extinction or variations in influent composition, such as nitrogen and organic matter loads, or the presence of toxic compounds [47]. The strong stochastic assembly of microbial communities has been observed in aquatic ecosystems, such as rivers [48] and reservoirs [49]. Similarly, deterministic processes play a critical role in shaping the microbial variation in water and sediment environments in lakes [50,51,52]. Our results show that the addition of microalgae affects the bacterial community structure, which in turn affects bacterial community assembly (Figure 5). This finding is similar to that of previous studies: the addition of microalgae triggers directional regulation of the bacterial community, which in turn affects the assembly of the bacterial community [32].

Community stability is an important indicator of the structure of a bacterial community. Research indicates that abundant keystone bacterial taxa, such as *Nitrospira* and *Gemmatimonas*, contribute to soil microbiome stability through specialized metabolic functions in “nitrogen metabolism” and “phosphonate and phosphinate metabolism” [31]. The stability of the soil bacterial community was further assessed using the average variation degree (AVD). AVD is calculated by measuring the deviation degree from the mean of the normally distributed ZOTU relative abundance among various treatment groups.

A lower AVD value indicates higher microbiome stability [31]. Deterministic processes in the bacterial community were more pronounced in water bodies where microalgae were added. Since microalgae increase the deterministic processes governing the bacterial community, they affect the stability of the community. Therefore, we hypothesize that microalgae would have an inhibitory or facilitative effect on the bacterial community through the metabolites they produce. This is similar to the findings of Ahmed A. Shibl et al. [53]. Results of the present study show that the addition of *T. weissflogii* resulted in a more stable bacterial community (Figure 6). *T. weissflogii* positively influences bacterial communities through strategies such as iron carriers and vitamin biosynthesis and exchange, leading to in situ reciprocity [54]. Results of the present study show that *T. weissflogii* had a positive effect on more bacteria than *N. oculata*, such as *Saprospiraceae*. In other words, *T. weissflogii* affects a wider variety of bacterial communities than *N. oculata.* In shrimp-farming waters, T. weissflogii inoculation stabilized the bacterial community more than the other treatments, but when *T. weissflogii* was combined with *N. oculata*, the two algae underwent community succession, resulting in community instability (Figure 6). This suggests that *T. weissflogii* is a key factor in the stability of the bacterial community. Coincidentally, it has been demonstrated that the diatom phylum dominates changes in bacterial community structure [55]. It is thus clear that the incorporation of *T. weissflogii* microalgae can control changes in the bacterial community and increase the deterministic processes governing it, thus improving the stability of the community, which is of great importance for healthy shrimp culture.

## 5. Conclusions

In the present study, different microalgae’s influence on bacterial community structure was examined through high-throughput sequencing technology. The results indicate that bacterial communities with different lifestyles had different response patterns to microalgae and nutrient salts, and microalgae indirectly affected bacterial communities through interactions with environmental factors. The present study found that the dynamic patterns of PA bacteria and FL bacteria were similar, but they differed significantly in terms of composition, representative bacteria, and drivers. Inoculation of microalgae could promote the aggregation of bacterial communities in shrimp aquaculture water, but its effect varied according to species. In conclusion, the present study revealed that microalgae and nutrient salts play an important role in the assembly and response mechanisms of microbial communities in shrimp aquaculture ecosystems. Adding specific microalgae to control the process of community change can improve the stability of the community and promote the healthy culture of shrimp. However, the current study has inherent limitations that preclude a comprehensive elucidation of the bacterial community dynamics, necessitating further research. For example, the relatively modest scale of the experiment may make it difficult to capture the overall picture of microbial community structure. Furthermore, real shrimp-aquaculture-rearing water, exposed to natural variations and anthropogenic factors, may introduce additional complexity not entirely replicated in our controlled setting. Moreover, the sensitivity of detecting low-abundance species might be compromised, and there exists a dearth of information regarding the function and metabolism of certain bacteria.

Further biochemical and molecular studies could reveal the mechanisms of interactions between microalgal and bacterial communities, including signaling molecules, release of metabolites, and possible ecological niche occupation involved.

Longer-timespan studies are conducted to better understand the long-term effects of microalgae on bacterial communities. This is critical to unraveling the dynamics of microalgae–bacteria interactions and the long-term response of ecosystems.

## Figures and Tables

**Figure 1 biology-13-00054-f001:**
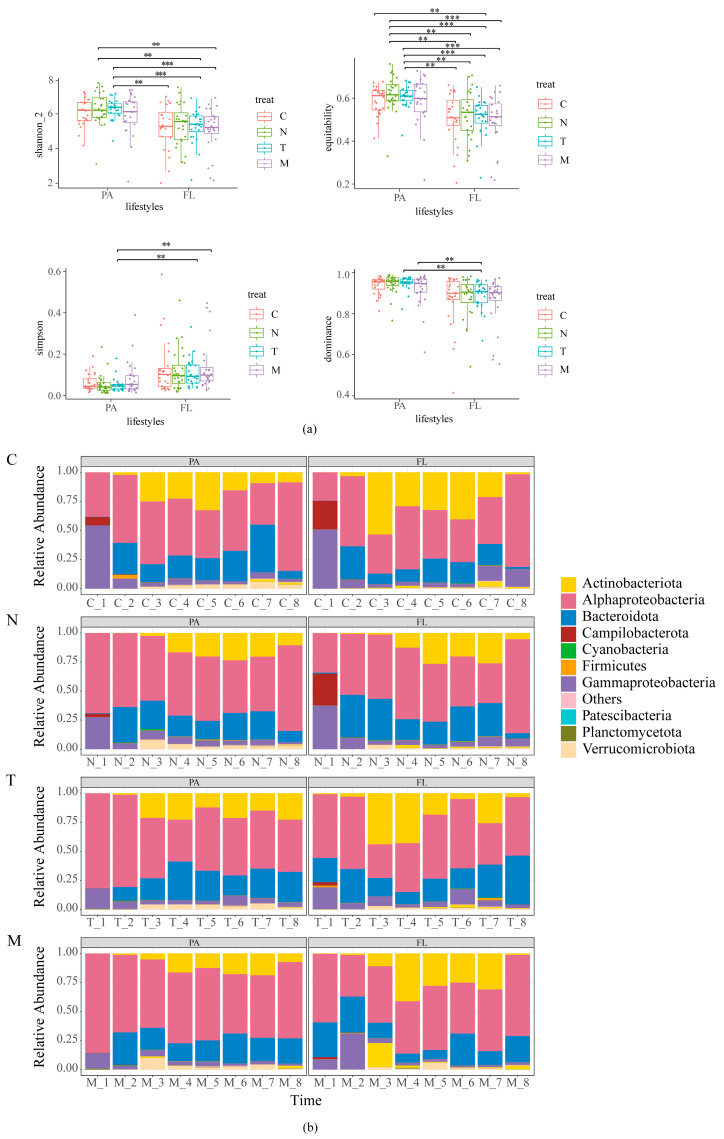
(**a**) Alpha diversity indices of particle-attached (PA) and free-living (FL) bacterial communities in the four groups (** *p* < 0.01, *** *p* < 0.001); (**b**) relative abundance of high-rank bacterial taxa under different lifestyles. Time refers to the number of days the experiment was conducted.

**Figure 2 biology-13-00054-f002:**
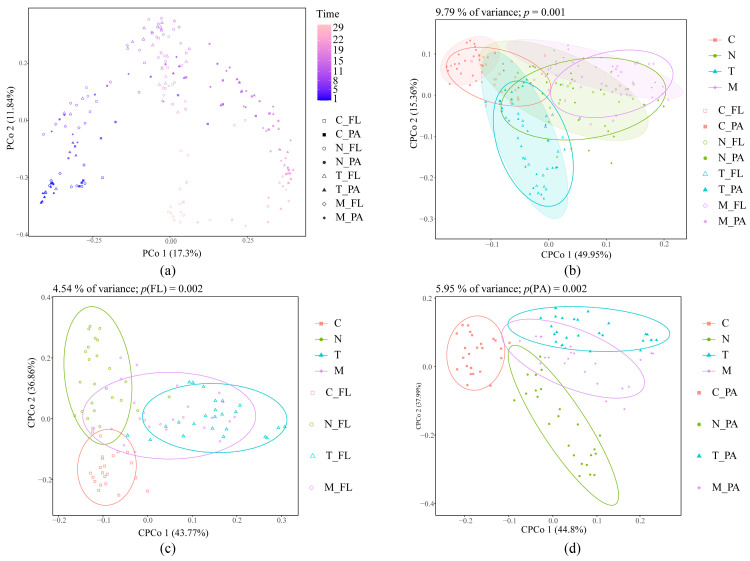
(**a**) Principal coordinate analysis (PCoA) based on the Bray–Curtis dissimilarities of whole bacterial communities; constrained PCoA (CPCoA) based on the Bray–Curtis dissimilarities of whole (**b**), FL (**c**), and PA (**d**) bacterial communities. Time refers to the number of days the experiment was conducted.

**Figure 3 biology-13-00054-f003:**
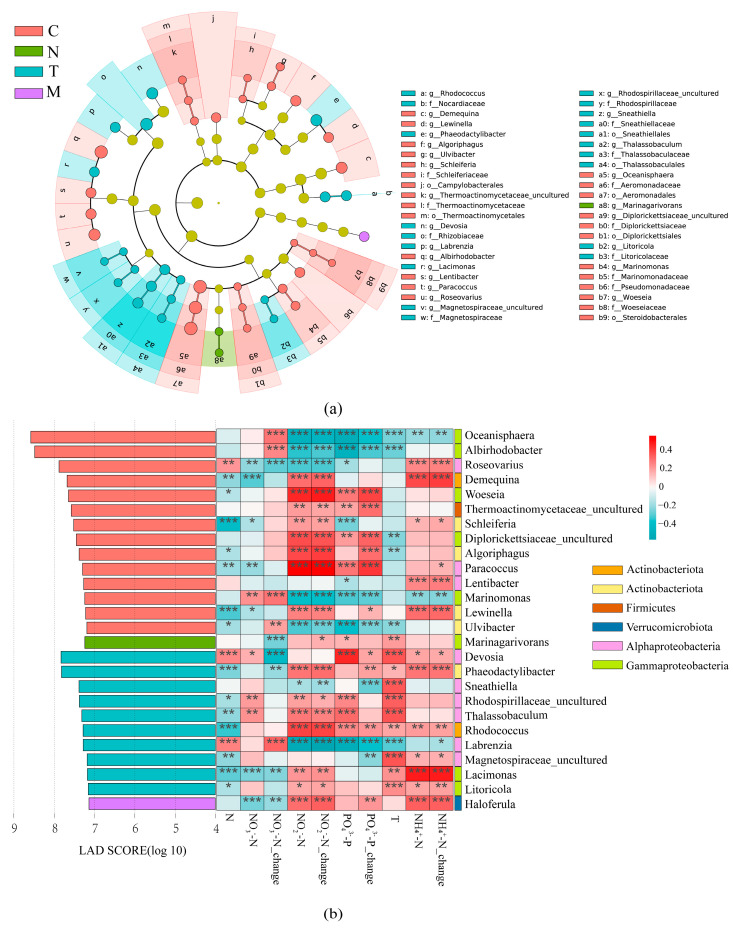
(**a**) Evolutionary branching diagram for LEfSe analysis of microbial communities (α > 5.5); (**b**) microbial community LDA and heatmap between key bacteria and environmental factors (genus level) (* *p* < 0.05, ** *p* < 0.01 and *** *p* < 0.001). Next to it is the phylum corresponding to the genus (Proteobacteria is divided into classes).

**Figure 4 biology-13-00054-f004:**
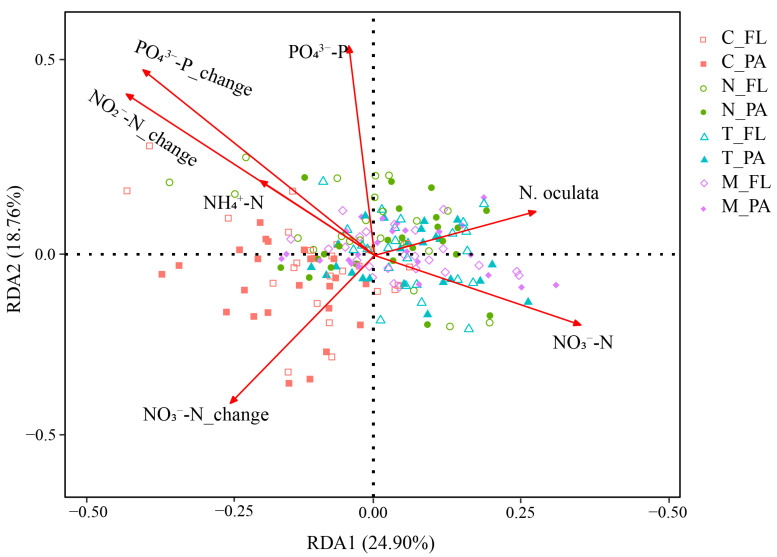
RDA of the influence of environmental factors on the bacterial community.

**Figure 5 biology-13-00054-f005:**
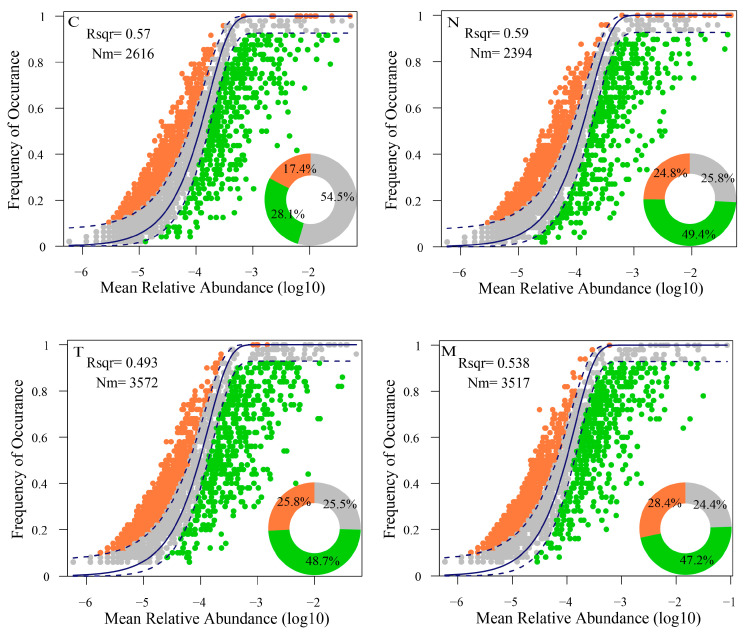
Neutral model analysis of bacterial communities in the water. Note: ZOTUs occurring more frequently than predicted were shown in orange, while those occurring less frequently are in green. The dark blue dashed line represents the 95% confidence interval around the model prediction. ZOTUs within the interval (gray) were considered neutrally distributed. The Rsqr value gauged the goodness-of-fit of the neutral model, ranging from 0 (no fit) to 1 (perfect fit). Estimated mobility (Nm) served as a measure of dispersal limitation.

**Figure 6 biology-13-00054-f006:**
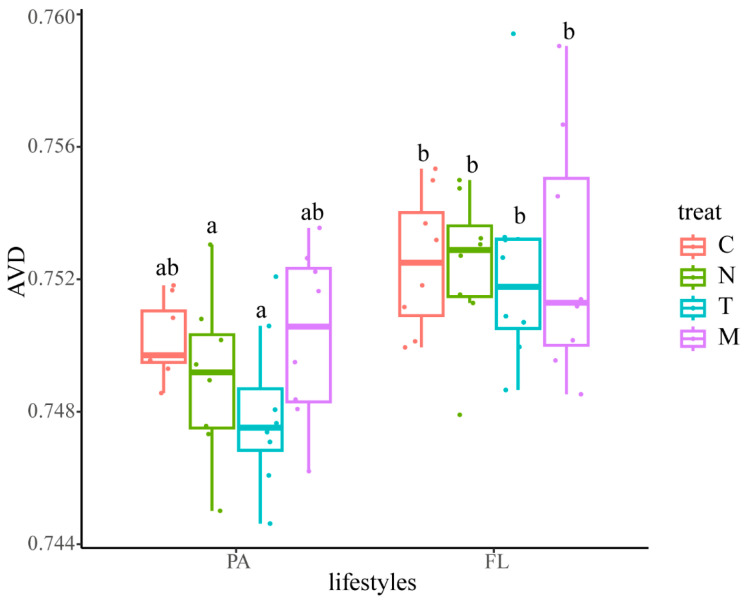
AVD analysis of environmental factors and microbial community stability. Different letters in the figure indicate significant differences (*p* < 0.05).

**Table 1 biology-13-00054-t001:** One-way ANOVA results for the four groups based on the levels of nutrient salts and the biomass of microalgae.

Factors (mg/L)	C	N	T	M
*N. oculata*	-	24.367 ± 28.849	-	20.146 ± 18.320
*T. weissflogii*	-	-	128.679 ± 106.771 ^b^	5.2 ± 3.487 ^a^
NH_4_^+^-N	0.344 ± 0.361	0.248 ± 0.230	0.237 ± 0.229	0.254 ± 0.249
NO_2_^−^-N	0.148 ± 0.246	0.216 ± 0.401	0.165 ± 0.262	0.161 ± 0.302
NO_3_^−^-N	1.729 ± 1.978 ^a^	3.806 ± 4.946 ^ab^	5.431 ± 4.980 ^b^	3.945 ± 4.702 ^ab^
PO_4_^3−^-P	0.889 ± 1.142 ^a^	1.836 ± 1.239 ^b^	2.125 ± 1.108 ^b^	1.903 ± 0.962 ^b^
NH_4_^+^-N_change	0.311 ± 0.361	0.227 ± 0.229	0.212 ± 0.231	0.182 ± 0.253
NO_2_^−^-N_change	0.166 ± 0.247	0.190 ± 0.400	0.142 ± 0.260	0.132 ± 0.303
NO_3_^−^-N_change	−3.104 ± 2.128 ^c^	−9.727 ± 5.233 ^a^	−6.448 ± 4.971 ^a^	−7.211 ± 5.411 ^b^
PO_4_^3−^-P_change	0.856 ± 1.145 ^ab^	1.491 ± 1.234 ^b^	0.531 ± 1.122 ^a^	0.676 ± 0.992 ^a^

Note: One-way ANOVA was initially conducted to identify the significant differences among groups. Rows without any letters in the data indicate no significant difference (*p* > 0.05). Subsequently, Tukey’s multiple comparisons were employed for post hoc test. Data within the same row with different letters indicate a significant difference (*p* < 0.05). C represents the control group, N represents the group with the addition of *N. oculata*, T represents the group with the addition of *T. weissflogii*, and M represents the group with the addition of the mixture of *N. oculata* and *T. weissflogii*. *N. oculata* represents the biomass of *N. oculata* and *T. weissflogii* represents the biomass of *T. weissflogii*. NH_4_^+^-N represents the concentration of ammonia nitrogen, NO_2_^−^-N represents the concentration of nitrite nitrogen, NO_3_^−^-N represents the concentration of nitrate nitrogen and PO_4_^3−^-P represents the concentration of reactive phosphate. NH_4_^+^-N_change, NO_2_^−^-N_change, NO_3_^−^-N_change, and PO_4_^3−^-P_change indicate the concentration of the amount of change in environmental factors obtained by subtracting the initial concentration.

**Table 2 biology-13-00054-t002:** PERMANOVA test for the effect of time, treatment, and lifestyles on changes in the bacterial community.

Factors	R^2^
treat	0.04394 ***
time	0.13181 ***
lifestyles	0.07824 ***
treat: time	0.03771 ***
treat: lifestyles	0.01214
time: lifestyles	0.02561 ***
treat: time: lifestyles	0.00933

Note: R^2^ indicates the contribution of individual variables and interactions between variables to driving changes in bacterial community structure (*** *p* < 0.001).

**Table 3 biology-13-00054-t003:** Mantel test of nutrient factors on bacterial community structure.

	ρ	F	*p*	ρ (PA)	ρ (FL)
*N. oculata*	0.0418	1.7796	0.003	0.1156	0.0273
NO_3_^−^-N	0.0761	1.7734	0.003	0.2147	0.0446
PO_4_^3−^-P	0.0505	2.7295	0.001	0.3125	0.2125
NH_4_^+^-N	0.0378	1.5658	0.010	0.0619	0.0263
PO_4_^3−^-P_change	0.0368	2.4080	0.001	0.2952	0.1774
NO_2_^−^-N_change	0.0274	1.3424	0.037	0.2693	0.1826
NO_3_^−^-N_change	0.0269	1.7725	0.001	0.1122	0.0281

Note: ρ indicates the contribution of individual variables and their interactions to driving changes in bacterial community structure.

## Data Availability

Data are contained within the article and Appendix A. The sequenced paired-end reads were deposited in the NCBI Sequence Read Archive with the BioProject number PRJNA1029339 and the accession number SRP467045 (https://www.ncbi.nlm.nih.gov/bioproject/PRJNA1029339).

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
