# Peer review of "Microalgae Inoculation Significantly Shapes the Structure, Alters the Assembly Process, and Enhances the Stability of Bacterial Communities in Shrimp-Rearing Water"

_biology, 2024, doi:10.3390/biology13010054_

Round 1

Reviewer 1 Report

Comments and Suggestions for Authors

Reviewer(s)' Comments to Author: 

Inoculation of microalgae into shrimp culture water contributes to the sustainable development of shrimp aquaculture and ecosystem conservation. The authors investigated the effect of inoculation of microalgae (Nannochloropsis oculata and Thalassiosira weissflogii) on the structure and assembly process of free-living and particle-associated bacterial communities in shrimp-culture waters over a 30-day period.

 The main problems of this work are the lack of data on changes in the abundance and biomass of microalgae during cultivation, so it is not clear on what basis the authors assessed the influence of microalgae and bacteria. The content of most of the articles cited in the introduction and discussion is inconsistent with the context in which they are used in this article. Therefore, the introduction and discussion need to be rewritten. Below are comments on the text.

Simple Summary

Line 15. “farnming” should be changed to “farming”

Line 17. What does "longitudinal study" mean?

1. Introduction

Many of the references (links) in the introduction are misused and do not fit the context in which they are used.

Line 59-61. This sentence focuses on bacteria: “In shrimp rearing ecosystems, bacteria, which are also present in various terrestrial and aquatic ecosystems [4], play a pivotal role in energy flows and nutrient cycling, ensuring a healthy and stable water microenvironment [5].”

[4] – “Amin, S.A. et al. Interaction and Signalling between a Cosmopolitan Phytoplankton and Associated Bacteria.” Although the introduction of this article contains the phrases “In terrestrial systems” and “In aquatic systems”, this article does not study terrestrial and aquatic systems, but the study of signaling between diatoms and bacteria during their laboratory co-cultivation.

And here is the link [5] – “Lightner, D.V. Virus Diseases of Farmed Shrimp…” about virus disease, not bacteria.

Could you provide a more relevant link?

For example, you can use in this context the article Azam, F., Malfatti, F. Microbial structuring of marine ecosystems. Nat Rev Microbiol 5, 782–791 (2007).

This article examines the role of bacteria in aquatic ecosystems.

Line 67-68. “Among the array of improvement strategies, inoculation with microalgae stands out as a widely employed strategy for bacterial manipulation in practical production [7].”

Please clarify this sentence. What does this phrase “inoculation with microalgae stands out as a widely employed strategy for bacterial manipulation in practical production” mean? Does this mean that microalgae inoculation is used to manipulate bacteria? But the article you cite [7] is Martínez-Ruiz et al., Micro-Algae Assisted Green Bioremediation of Water Pollutants Rich Leachate and Source Products Recovery. – explores bioremediation mainly with algae only. Bacteria are mentioned only occasionally.

Line 71-72. “the other involves microalgae shaping the bacterial composition by altering the nutrient structure of the water column [9].” Water column of what? [9] – Mujtaba et al. Advanced Treatment of Wastewater Using Symbiotic Co-Culture of Microalgae and Bacteria.

– this article is about wastewater. This article does not mention the phrase “water column”.

Line 75-76.

“However, the comparable and contrasting impacts of microalgae on FL and PA bacteria remain understudied.” Why didn't you quote articles on the effect of microalgae on FL and PA bacteria? There are a large number of articles on the influence of phytoplankton (microalgae) bloom on FL and PA bacteria in marine and freshwater ecosystems. In addition, there are works on the influence of microalgae on FL and PA bacteria when they are co-cultivated (for example, Bagatini, et al. (2014). Host-specificity and dynamics in bacterial communities associated with bloom-forming freshwater phytoplankton. PloS one, 9(1 ), e85950.).

Line 78-80. “FL bacteria, being attached to organic particles, primarily contribute to the degradation of organic matter and the elimination of nitrogenous waste [12].”

[12] - Mingming, et al., Phytoplankton and Bacterioplankton Abundances and Community Dynamics in Lake Erhai. This article is about seasonal dynamics of phytoplankton and bacterioplankton in Lake Erhai, not about “degradation of organic matter and the elimination of nitrogenous waste”

Line 80-81. “In contrast, PA bacteria engage in activities such as surface colonization, protection against predation and the improved acquisition of nutrients [13].”

[13] – Battin et al. The Ecology and Biogeochemistry of Stream Biofilms. – This article does not use the phrase “particle-attached bacteria”.

Line 86-87. “Bacterial growth is heavily reliant on the nutrient structure, with distinct bacterial growth patterns being directed by specific nutrient factors [14].” could you write this sentence more clearly?

Line 87-88. “A stable water quality environment is essential for the assembly and maintenance of rearing-water bacterial communities [15].”

15. Yang, L. et al. Spatio-Temporal Distribution, Photoreactivity and Environmental Control of Dissolved Organic Matter in the Sea-Surface Microlayer of the Eastern Marginal Seas of China.  – this article is about dissolved organic matter in the sea-surface microlayer of the eastern marginal seas of China. In addition, it counts the number of bacteria, but does not estimate the composition of bacterial communities, and nothing is said about their assembly. This article does not use the phrase “rearing-water bacterial communities.”

Line 92-93. “Nannochloropsis Oculata and Thalassiosira weissflogii” should be changed to “eustigmatophyte Nannochloropsis oculata and diatom Thalassiosira weissflogii”

Line 95. “composition and structure” – What is the difference between composition and structure? It may be better to use only one of these terms, since they are used in different articles to describe the same patterns.

2. Materials and methods

2.1. Microalgal culture

Line 106-107: Were the N. oculata and T. weissflogii cultures axenic?

2.2. Experimental design

Line 123-124: “(with the addition of N. oculata and T. weissflogii)” In what ratio was the mixture of these species used?

Line 128-129: Was the experiment carried out under natural day and night conditions or did additional lighting be used? Was the cultivation outdoors or indoors (not clear from the Experimental design description)? Was mixing done in the tanks?

2.3. Environmental and bacterial sample collection

Line 141-142. “In total, 36 (4 groups × 3 replicates × 8 time points) PA bacterial samples and 36 (4 groups × 3 replicates × 8 time points) FL bacterial samples were collected” Why in total, 36? 4 Ñ… 3 Ñ… 8 = 96.

Line 145. “NH4+-N, NO3--N, NO2--N, and PO43-P” What do “-N” and “-P” mean and why are they needed?

2.4. Bacterial Illumina HiSeq sequencing and bioinformatic analysis

Line 167. https://www.ncbi.nlm.nih.gov/sra/PRJNA1029339 When I go to this link, a record is displayed “The requested page does not exist.”

Line 169-170. “unoise_alpha = 2 and minsize = 4 as per default settings” What commands in Usearch use these parameters and what do they do?

“using the UNOISE3 pipeline” In Usearch, to obtain zOTU, it is not the UNOISE3 pipeline that is used, but only the command “-unoise3” (algorithm). All other commands are the same as for UPARSE (when receiving OTU). Therefore, “using the UNOISE3 pipeline” can be replaced with “using the UNOISE3” or “using the UNOISE3 algorithm”.

Line 171: Did you remove singletons, zOTUs of chloroplasts from the data? If yes, please indicate this. Bacterial communities from phytoplankton samples tend to amplify the chloroplasts (classified as Cyanobacteria) of the phytoplankton, but you have very low proportions of Cyanobacteria in the samples.

3. Results

Line 211: “Regarding the impact of microalgal inoculation, T. weissflogii decreased and N. oculata remained stable in the combined microalgal inoculation treatment.” What declined and what remained stable? Number or biomass? This article lacks data on the dynamics of microalgae when cultivated in these tanks.

Line 214: What do NH4+-N_change, NO2--N_change, NO3--N_change, PO43--P_change mean?

 Line 217: Table S1:

There are four groups in your experiment: C (Control), N (with N. oculata), T (with T. weissflogii), and Group M (with N. oculata and T. weissflogii) (listed in methods).

What does “W” mean in Table S1?

Table S1 is called “Significance of alpha-diversity index between different lifestyles.” Does this table only show significance or does it show diversity index values? What values are shown in this table? Are these the average values from three replicates of the experiment? Did you average all samples from 8 time points? Why is it not written what was done with 192 samples, so that the resulting diversity values for 8 lines were obtained? Also, add a column with the number of reads.

Line 228: “in the three groups” – Does Fig.1a show three or four groups?

Line 228: “Bacterial abundance stacking” – maybe you need to write more clearly: “Relative abundance of high-rank bacterial taxa”

Line 241: Figure 2a: What does the blue-pink scale on the right show? Add a signature.

Line 261: Figure 3: You could write the abbreviated name of the major taxon (phylum or class) next to the genus names to make it clearer. Or choose some other way to indicate which taxa these genera belong to.

Line 276: “mantel” should be changed to “Mantel”

Line 289-290: “A neutral model-based analysis at the family level was conducted in the upper, middle, and lower sections” – What are upper, middle, and lower sections? These are shown in Figure S1 but it is not clear what they mean.

Line 290: Figure S1: In supplementary figure, this is what the indication of the authors of your article looks like: “Firstname Lastname 1, Firstname Lastname 2 and Firstname Lastname 2,*”. Tidy up the supplementary figure.

“Figure S1. Taxonomic distribution of OTUs (at the family level) for the four groups.” – Write a clearer description of what this picture shows. What does Above, Neutral, Below mean?

4. Discussion

Line 316-318: “Our findings revealed that N. oculata exerted a more pronounced influence on nutrient levels than T. weissflogii and the combination of the two species.” – In your article there is no data on changes in the number and biomass of microalgae during the experiment. It is not known whether the T. weissflogii culture continued to grow, or whether it died after inoculation, so there was a low impact on bacterial communities.

Line 325-326: Microalgae exert a strong influence on the diversity and composition of bacterial  communities, as indicated in previous studies [23].

23. Datta, M.S.; Sliwerska, E.; Gore, J.; Polz, M.F.; Cordero, O.X. Microbial Interactions Lead to Rapid Micro-Scale Successions on Model Marine Particles. – this article is about bacterial colonization of particle organic matter. I checked through a search and did not find the word “microalgae” in this article.

Line 332-336: “This finding aligns with previous observations where microalgae, specifically N. oculata, influenced the intestinal bacterial community of European seabass. N. oculata increased the representation of Bacillus, known for their probiotic potential, and reduced the abundance of potentially pathogenic bacteria such as Acinetobacter [24].

24. Ferreira et al. Gracilaria Gracilis and Nannochloropsis Oceanica, Singly or in Combination, in Diets Alter the Intestinal Microbiota of European Seabass (Dicentrarchus Labrax). – this article is about Nannochloropsis oceanica, not N. oculata, as you write.

Line 337-338: Chlorophyta influence bacterial communities through metabolites [4].

4. Amin et al. Interaction and Signalling between a Cosmopolitan Phytoplankton and Associated Bacteria. – this article is about interaction between diatom and bacteria. This article does not examine the influence of Chlorophyta on bacterial communities.

Line 342-343: In microalgae-inoculated waters, T. weissflogii was symbiotic with Marivita [26].

What is the reason for the emphasis on Marivita? It was not shown in your work, but in the article “26. Sun et al. Diatom Red Tide Significantly Drives the Changes of Microbiome in Mariculture Ecosystem”, in addition to this genus, there were several more genera of symbiotic bacteria.

Line 347-350: “When the nitrogen source is ammonium, the oxygen generated by microalgae has a positive influence on nitrifying bacteria [27]. Conversely, when nitrate serves as the nitrogen source, oxygen can have an inhibitory effect on denitrifying bacteria [27].”

In your work, have nitrifying and denitrifying bacteria been identified in bacterial communities? If not, then why discuss them.

Line 354-357: In a lake ecosystem, the FL bacterial community played a critical role in methane utilization, while the PA community seemed to have contributed more to biogeochemical cycling [31].

31. Shen, et al. Similar Assembly Mechanisms but Distinct Co-Occurrence Patterns of Free-Living vs. Particle-Attached Bacterial Communities across Different Habitats and Seasons in Shallow, Eutrophic Lake Taihu.

In this work, it was in Lake Taihu that the FL bacterial community played a critical role in methane utilization, but in other lakes FL bacteria may play other roles, so it probably should not be generalized to “a lake ecosystem.” Also, in Shen, et al. it is written that “the PA bacteria contributed more to biogeochemical cycling of carbon.”, but you write “the PA community seemed to have contributed more to biogeochemical cycling.” It is necessary to cite literature sources more accurately and correctly.

Line 361-362: “PA bacteria have richer bacterial communities than FL bacteria” What does “richer” mean?

Line 374-375: “Deterministic processes are considered the main drivers shaping microecological communities in aquatic environments [35].” What is “microecological communities”?

35. Cruaud, et al. Annual Protist Community Dynamics in a Freshwater Ecosystem Undergoing Contrasted Climatic Conditions: The Saint-Charles River (Canada). – This article is about protist (microeukaryotic) communities in river. The phrases “Deterministic processes” and “microecological communities” are missing from this article.

Line 375-378: “It has been shown that stochastic processes dominate pathogenic community assembly, while deterministic ecological niche processes are the dominant mechanisms controlling pathogenic community assembly [36].” – This sentence was taken out of context and therefore lost its meaning.

36. Zhang, et al. Differences in Pathogenic Community Assembly Processes and Their Interactions with Bacterial Communities in River and Lake Ecosystems – The original sentence looks like this: “Stochastic processes dominate pathogenic community assembly in riverine habitats, while deterministic ecological niche processes are the dominant mechanisms controlling pathogenic community assembly in the lake habitat.”

Line 403-404: “Our results show that T. weissflogii had a positive effect on more bacteria than Microcystis aeruginosa, such as Saprospiraceae.” What does this mean? Your work does not evaluate the effect of Microcystis aeruginosa on bacteria, including Saprospiraceae.

Line 409-411: “Coincidentally, it has been demonstrated that the diatom phylum dominates changes in bacterial community structure [43].” again incorrect quotation.

43. Klamt, et al. An Extreme Drought Event Homogenises the Diatom Composition of Two Shallow Lakes in Southwest China. – this article is only about diatoms. It does not study the effect of diatoms on bacteria. This article doesn't even contain the phrase “bacterial community.”

Comments on the Quality of English Language

Moderate editing of English language required

Author Response

We are very grateful for your affirmation of our work. We greatly appreciate your comments and the thorough revision that greatly contributed to the improvement of our manuscript. Please find the detailed responses below. Red color indicated the corresponding revisions in the re-submitted files.

Comments 1: Inoculation of microalgae into shrimp culture water contributes to the sustainable development of shrimp aquaculture and ecosystem conservation. The authors investigated the effect of inoculation of microalgae (Nannochloropsis oculata and Thalassiosira weissflogii) on the structure and assembly process of free-living and particle-associated bacterial communities in shrimp-culture waters over a 30-day period. The main problems of this work are the lack of data on changes in the abundance and biomass of microalgae during cultivation, so it is not clear on what basis the authors assessed the influence of microalgae and bacteria. The content of most of the articles cited in the introduction and discussion is inconsistent with the context in which they are used in this article. Therefore, the introduction and discussion need to be rewritten. Below are comments on the text.

Responses 1: We sincerely appreciate your valuable comments and the meticulous revisions, which have significantly enhanced the quality of our manuscript. Notably, we have incorporated the data on microalgae biomass changes into Figure S1, as elaborated in Response 23. Moreover, we have modified the improper references, and you can find the specifics of these corrections in related Responses. The red color highlights these corresponding revisions in the resubmitted files.

Comments 2: Line 15. “farnming” should be changed to “farming”

Responses 2: Thank you for bringing this to our attention. The term " farnming " has been corrected to "farming ", as can been seen in the revised manuscript in Line 15.

Comments 3: Line 17. What does "longitudinal study" mean?

Responses 3: "Longitudinal study" refers to a research design characterized by multiple observations or measurements of the same group or individual over an extended period. This approach enables a comprehensive understanding of how microbial communities success and change over time. Faust et al. use the term "longitudinal study" to denote the study of patterns of change in a microbial community over a longitudinal time series [1]. In addition, Flores et al. used a longitudinal study of a large number of healthy and diseased hosts to identify the ecological factors that shape the diversity, composition, and dynamics of the human microbiome [2].

References:

1. Faust, K.; Lahti, L.; Gonze, D.; de Vos, W.M.; Raes, J. Metagenomics Meets Time Series Analysis: Unraveling Microbial Community Dynamics. Current Opinion in Microbiology 2015, 25, 56–66, doi:10.1016/j.mib.2015.04.004.

2. Flores, G.E.; Caporaso, J.G.; Henley, J.B.; Rideout, J.R.; Domogala, D.; Chase, J.; Leff, J.W.; Vázquez-Baeza, Y.; Gonzalez, A.; Knight, R.; et al. Temporal Variability Is a Personalized Feature of the Human Microbiome. Genome Biol 2014, 15, 531, doi:10.1186/s13059-014-0531-y.

1. Introduction

Comments 4: Many of the references (links) in the introduction are misused and do not fit the context in which they are used.

Responses 4: We appreciate your attention to the references in the introduction and recognize the significance of ensuring their accuracy and contextual relevance. We have promptly reviewed and revised the references to ensure they align appropriately with the context of our study. If you have any further suggestions or concerns, please don't hesitate to let us know. Your input is highly valuable, and we are committed to addressing any remaining issues to enhance the overall quality of our manuscript.

Comments 5: Line 59-61. This sentence focuses on bacteria: “In shrimp rearing ecosystems, bacteria, which are also present in various terrestrial and aquatic ecosystems [4], play a pivotal role in energy flows and nutrient cycling, ensuring a healthy and stable water microenvironment [5].”

[4] – “Amin, S.A. et al. Interaction and Signalling between a Cosmopolitan Phytoplankton and Associated Bacteria.” Although the introduction of this article contains the phrases “In terrestrial systems” and “In aquatic systems”, this article does not study terrestrial and aquatic systems, but the study of signaling between diatoms and bacteria during their laboratory co-cultivation.

And here is the link [5] – “Lightner, D.V. Virus Diseases of Farmed Shrimp…” about virus disease, not bacteria.

Could you provide a more relevant link?

For example, you can use in this context the article Azam, F., Malfatti, F. Microbial structuring of marine ecosystems. Nat Rev Microbiol 5, 782–791 (2007).

This article examines the role of bacteria in aquatic ecosystems.

Responses 5: After carefully consideration of your observations regarding the references in the specified sentence, we concur that the originally cited references [4] and [5] may not be the most appropriate for supporting this argument. Consequently, we have replaced them with the article you recommended [6] (6.     Azam, F.; Malfatti, F. Microbial Structuring of Marine Ecosystems. Nat Rev Microbiol 2007, 5, 782–791, doi:10.1038/nrmicro1747.) and a reference related to the role of bacteria in terrestrial systems [5] (5.        Van Der Heijden, M.G.A.; Bardgett, R.D.; Van Straalen, N.M. The Unseen Majority: Soil Microbes as Drivers of Plant Diversity and Productivity in Terrestrial Ecosystems. Ecology Letters 2008, 11, 296–310, doi:10.1111/j.1461-0248.2007.01139.x.), as can been seen in the revised manuscript in Line 64 and Line 518-522.

Comments 6: Line 67-68. “Among the array of improvement strategies, inoculation with microalgae stands out as a widely employed strategy for bacterial manipulation in practical production [7].”

Please clarify this sentence. What does this phrase “inoculation with microalgae stands out as a widely employed strategy for bacterial manipulation in practical production” mean? Does this mean that microalgae inoculation is used to manipulate bacteria? But the article you cite [7] is Martínez-Ruiz et al., Micro-Algae Assisted Green Bioremediation of Water Pollutants Rich Leachate and Source Products Recovery. – explores bioremediation mainly with algae only. Bacteria are mentioned only occasionally.

Responses 6: The implication of this statement is that the introduction of microalgae into a shrimp rearing system is a widely adopted strategy used to manipulate bacterial community structure. To avoid ambiguity, we have revised the sentence to "Among the array of improvement strategies, inoculation with microalgae stands out as a widely employed strategy for manipulating bacterial community structure in practical production." Additionally, the reference has been replaced with [8] (8.   Heyse, J.; Props, R.; Kongnuan, P.; De Schryver, P.; Rombaut, G.; Defoirdt, T.; Boon, N. Rearing Water Microbiomes in White Leg Shrimp (  LITOPENAEUS VANNAMEI  ) Larviculture Assemble Stochastically and Are Influenced by the Microbiomes of Live Feed Products. Environmental Microbiology 2021, 23, 281–298, doi:10.1111/1462-2920.15310.). The revisions can be seen in the revised manuscript in Line 71 and Line 525-528.

Comments 7: Line 71-72. “the other involves microalgae shaping the bacterial composition by altering the nutrient structure of the water column [9].” Water column of what? [9] – Mujtaba et al. Advanced Treatment of Wastewater Using Symbiotic Co-Culture of Microalgae and Bacteria.

– this article is about wastewater. This article does not mention the phrase “water column”.

Responses 7: The term “water column” has been changed to “water’, as can be seen in Line 74.

Comments 8: Line 75-76.

“However, the comparable and contrasting impacts of microalgae on FL and PA bacteria remain understudied.” Why didn't you quote articles on the effect of microalgae on FL and PA bacteria? There are a large number of articles on the influence of phytoplankton (microalgae) bloom on FL and PA bacteria in marine and freshwater ecosystems. In addition, there are works on the influence of microalgae on FL and PA bacteria when they are co-cultivated (for example, Bagatini, et al. (2014). Host-specificity and dynamics in bacterial communities associated with bloom-forming freshwater phytoplankton. PloS one, 9(1 ), e85950.).

Responses 8: The exaggerated expressions and inappropriate references have been modified in the revised manuscript, Line 77-80 and Line 541-549. Added references [13-15] to the manuscript (13.     Park, B.S.; Choi, W.-J.; Guo, R.; Kim, H.; Ki, J.-S. Changes in Free-Living and Particle-Associated Bacterial Communities Depending on the Growth Phases of Marine Green Algae, Tetraselmis Suecica. Journal of Marine Science and Engineering 2021, 9, 171, doi:10.3390/jmse9020171. 14. Bagatini, I.L.; Eiler, A.; Bertilsson, S.; Klaveness, D.; Tessarolli, L.P.; Vieira, A.A.H. Host-Specificity and Dynamics in Bacterial Communities Associated with Bloom-Forming Freshwater Phytoplankton. PLOS ONE 2014, 9, e85950. 15. Yang, C.; Wang, Q.; Simon, P.N.; Liu, J.; Liu, L.; Dai, X.; Zhang, X.; Kuang, J.; Igarashi, Y.; Pan, X.; et al. Distinct Network Interactions in Particle-Associated and Free-Living Bacterial Communities during a Microcystis Aeruginosa Bloom in a Plateau Lake. Frontiers in Microbiology 2017, 8, 1201, doi:10.3389/fmicb.2017.01202.)

Comments 9: Line 78-80. “FL bacteria, being attached to organic particles, primarily contribute to the degradation of organic matter and the elimination of nitrogenous waste [12].”

[12] - Mingming, et al., Phytoplankton and Bacterioplankton Abundances and Community Dynamics in Lake Erhai. This article is about seasonal dynamics of phytoplankton and bacterioplankton in Lake Erhai, not about “degradation of organic matter and the elimination of nitrogenous waste”

Responses 9: The inappropriate reference has replaced with [18] (18.    Jain, A.; Krishnan, K.P. Differences in Free-Living and Particle-Associated Bacterial Communities and Their Spatial Variation in Kongsfjorden, Arctic. Journal of Basic Microbiology 2017, 57, 827–838, doi:10.1002/jobm.201700216..), as can be seen in Line 84 and Line 556-558.

Comments 10: Line 80-81. “In contrast, PA bacteria engage in activities such as surface colonization, protection against predation and the improved acquisition of nutrients [13].”

[13] – Battin et al. The Ecology and Biogeochemistry of Stream Biofilms. – This article does not use the phrase “particle-attached bacteria”.

Responses 10: The inappropriate reference has replaced with [17, 19, 20] (17.       Jiangtao L.; Bingbing W.; Jiani W.; Ying L.; Shamik D.; Li Z.; Jiasong F. Variation in abundance and community structure of particle-attached and free-living bacteria in the South China Sea. Deep-Sea Research 2015, 122, 64–73, doi:10.1016/j.dsr2.2015.07.006. 19. Cai, X.; Yao, L.; Hu, Y.; Jiang, H.; Shen, M.; Hu, Q.; Wang, Z.; Dahlgren, R.A. Particle-Attached Microorganism Oxidation of Ammonia in a Hypereutrophic Urban River. Journal of Basic Microbiology 2019, 59, 511–524, doi:10.1002/jobm.201800599. 20.Tang, X.; Chao, J.; Gong, Y.; Wang, Y.; Wilhelm, S.W.; Gao, G. Spatiotemporal Dynamics of Bacterial Community Composition in Large Shallow Eutrophic Lake Taihu: High Overlap between Free-Living and Particle-Attached Assemblages. Limnology and Oceanography 2017, 62, 1366–1382, doi:10.1002/lno.10502.) in Line 86 and Line 553-564.

Comments 11: Line 86-87. “Bacterial growth is heavily reliant on the nutrient structure, with distinct bacterial growth patterns being directed by specific nutrient factors [14].” could you write this sentence more clearly?

Responses 11: We appreciate your attention to detail. Based on your suggestion, we have made changes in the manuscript to make it clearer, as can be seen in Line 91-93. We have cited reference [22] (22. Ebeling, J.M.; Timmons, M.B.; Bisogni, J.J. Engineering Analysis of the Stoichiometry of Photoautotrophic, Autotrophic, and Heterotrophic Removal of Ammonia–Nitrogen in Aquaculture Systems. Aquaculture 2006, 257, 346–358, doi:10.1016/j.aquaculture.2006.03.019.), Line 568-570.

Comments 12: Line 87-88. “A stable water quality environment is essential for the assembly and maintenance of rearing-water bacterial communities [15].” 15. Yang, L. et al. Spatio-Temporal Distribution, Photoreactivity and Environmental Control of Dissolved Organic Matter in the Sea-Surface Microlayer of the Eastern Marginal Seas of China. – this article is about dissolved organic matter in the sea-surface microlayer of the eastern marginal seas of China. In addition, it counts the number of bacteria, but does not estimate the composition of bacterial communities, and nothing is said about their assembly. This article does not use the phrase “rearing-water bacterial communities.”

Responses 12: Thank you very much for your suggestions. We apologize for overlooking this issue. We have revised and replaced the correct reference in the manuscript [23] (23. Zheng, Y.; Yu, M.; Liu, J.; Qiao, Y.; Wang, L.; Li, Z.; Zhang, X.-H.; Yu, M. Bacterial Community Associated with Healthy and Diseased Pacific White Shrimp (Litopenaeus Vannamei) Larvae and Rearing Water across Different Growth Stages. Frontiers in Microbiology 2017, 8, 1362, doi:10.3389/fmicb.2017.01362.), Line 94-95 and Line 571-573.

Comments 13: Line 92-93. “Nannochloropsis Oculata and Thalassiosira weissflogii” should be changed to “eustigmatophyte Nannochloropsis oculata and diatom Thalassiosira weissflogii

Responses 13: The terms have been updated as per your suggestion, as can been seen in the revised manuscript in Line 100-101.

Comments 14: Line 95. “composition and structure” – What is the difference between composition and structure? It may be better to use only one of these terms, since they are used in different articles to describe the same patterns.

Responses 14: The term "composition" has been removed and retained "structure" as per your suggestion in Line 103.

2. Materials and methods

2.1. Microalgal culture

Comments 15: Line 106-107: Were the N. oculata and T. weissflogii cultures axenic?

Responses 15: Yes, N. oculata and T. weissflogii cultures are sterile. The microalgae were axenic when initially obtained from the Marine Biotechnology Laboratory. However, they were difficult to maintain absolute sterility when using 10-L plastic cylindrical photoreactors for productive culture in the production base, although the culture medium and facilities had been sterilized by autoclaving and hypochlorous acid (HOCl). To ensure the unavoidable contaminative bacteria in the microalgae suspension had no effect on subsequent experiment, we further conducted MRPP, ANOSIM and Adnois analyses on the bacterial community on the first day. The results showed that there was no significant difference among groups in both PA and FL bacterial communities (Table S2).

2.2. Experimental design

Comments 16: Line 123-124: “(with the addition of N. oculata and T. weissflogii)” In what ratio was the mixture of these species used?

Responses 16: Considering the difference in biological volume between N. oculata and T. weissflogii, we chose the microalgal biomass as the indicator for experiment design and results presentation instead of density. N. oculata concentration was 3 × 105 cells/mL approximately and T. weissflogii concentration was 8 × 103 cells/mL approximately. Therefore, the ratio of N. oculata to T. weissflogii in the mixed microalgae of group M is approximately 37.5. Relevant information has been added in the revised manuscript, Line 130-134.

Comments 17: Line 128-129: Was the experiment carried out under natural day and night conditions or did additional lighting be used? Was the cultivation outdoors or indoors (not clear from the Experimental design description)? Was mixing done in the tanks?

Responses 17: The experiment was conducted under natural day and night conditions and the shrimp culture was conducted indoors in the transparent glasses (Ningbo University production site in Ningbo, China). Two microalgae with equal biomass were mixed in tanks. Relevant information has been added in the revised manuscript, Line 138-141.

2.3. Environmental and bacterial sample collection

Comments 18: Line 141-142. “In total, 36 (4 groups × 3 replicates × 8 time points) PA bacterial samples and 36 (4 groups × 3 replicates × 8 time points) FL bacterial samples were collected” Why in total, 36? 4 Ñ… 3 Ñ… 8 = 96.

Responses 18: Thanks for the careful review. We apologize for the oversight, and the error in the calculation has been rectified in the manuscript, Line 147-148.

Comments 19: Line 145. “NH4+-N, NO3--N, NO2--N, and PO43--P” What do “-N” and “-P” mean and why are they needed?

Responses 19: “NH4+-N” refers to ammonium nitrogen. “NO3--N” refers to nitrate nitrogen. “NO2--N” refers to nitrite nitrogen. “PO43--P” refers to phosphate phosphorus. The “-N” and “-P” indicate that the concentration is expressed in terms of the specific element (nitrogen or phosphorus) rather than the entire compound.

2.4. Bacterial Illumina HiSeq sequencing and bioinformatic analysis

Comments 20: Line 167. https://www.ncbi.nlm.nih.gov/sra/PRJNA1029339 When I go to this link, a record is displayed “The requested page does not exist.”

Responses 20: The link (https://www.ncbi.nlm.nih.gov/bioproject/PRJNA1029339) has been updated, as can be seen in the manuscript, Line 190.

Comments 21: Line 169-170. “unoise_alpha = 2 and minsize = 4 as per default settings” What commands in Usearch use these parameters and what do they do?

“using the UNOISE3 pipeline” In Usearch, to obtain zOTU, it is not the UNOISE3 pipeline that is used, but only the command “-unoise3” (algorithm). All other commands are the same as for UPARSE (when receiving OTU). Therefore, “using the UNOISE3 pipeline” can be replaced with “using the UNOISE3” or “using the UNOISE3 algorithm”.

Responses 21: The “unoise_alpha” is a parameter used in the USEARCH, specifically in the “unoise3 algorithm”. Its purpose is to set the alpha value for the UNOISE algorithm. This alpha parameter controls the trade-off between sensitivity and specificity in detecting noise in sequence data. A higher alpha value increases sensitivity but may also increase the likelihood of including noisy sequences.

The “minsize” is another parameter used with unoise3 or unoise2, and its purpose is to set the minimum size (length) of the denoised sequences to be retained. Sequences shorter than this minimum size are discarded.

Both parameters aid in filtering out very short sequences that may result from sequencing errors or other artifacts. To enhance reader comprehension, relevant information has been incorporated in the revised manuscript, Line 191-193. Also, the term “using the UNOISE3 pipeline” has been replaced by “using the UNOISE3 algorithm”, as can be seen in Line 193.

Comments 22: Line 171: Did you remove singletons, zOTUs of chloroplasts from the data? If yes, please indicate this. Bacterial communities from phytoplankton samples tend to amplify the chloroplasts (classified as Cyanobacteria) of the phytoplankton, but you have very low proportions of Cyanobacteria in the samples.

Responses 22: Yes, we removed singletons, zOTUs of chloroplasts from the data. The low-quality reads, singletons and chimeric sequences were decarded during sequence analysis process prior to clustered ZOTUs. Therefore, the singletons were not considered in the subsequent diversity analysis. To prevent any misunderstandings and confusion, we have included relevant information in the revised manuscript, Line 196-199.

3. Results

Comments 23: Line 211: “Regarding the impact of microalgal inoculation, T. weissflogii decreased and N. oculata remained stable in the combined microalgal inoculation treatment.” What declined and what remained stable? Number or biomass? This article lacks data on the dynamics of microalgae when cultivated in these tanks.

Responses 23: Considering the significant different in the volume between T. weissflogii and N. oculata, we used biomass instead of abundance as an indicator to measure the dynamics of microalgae. In the Table 1 of the main manuscript, the mean and standard deviation of the changes in both microalgae were presented over time. The relevant information has been added in the revised manuscript, Line 244-245 and Line 247-248. Furthermore, the specific variations of the two microalgae are illustrated in Figure S1, which can be accessed in the supplementary material for download.

Comments 24: Line 214: What do NH4+-N_change, NO2--N_change, NO3--N_change, PO43--P_change mean?

Responses 24: NH4+-N_change, NO2--N_change, NO3--N_change and PO43--P_change indicate the concentration of the amount of change in environmental factors.

Comments 25: Line 217: Table S1:

There are four groups in your experiment: C (Control), N (with N. oculata), T (with T. weissflogii), and Group M (with N. oculata and T. weissflogii) (listed in methods).

What does “W” mean in Table S1?

Table S1 is called “Significance of alpha-diversity index between different lifestyles.” Does this table only show significance or does it show diversity index values? What values are shown in this table? Are these the average values from three replicates of the experiment? Did you average all samples from 8 time points? Why is it not written what was done with 192 samples, so that the resulting diversity values for 8 lines were obtained? Also, add a column with the number of reads.

Responses 25: The “W” error has been corrected to “T”. Table S1 displays the mean ± standard deviation of diversity index values and the significance between different groups and different lifestyles. This means that the 192 samples were categorized into 8 groups based on both group and lifestyle. Relevant information has been incorporated in the title and notes of Table S1.

Comments 26: Line 228: “in the three groups” – Does Fig.1a show three or four groups?

Responses 26: Thank you for your careful review. We are sorry that this was an oversight on our part. We have changed to four groups in Line 271.

Comments 27: Line 228: “Bacterial abundance stacking” – maybe you need to write more clearly: “Relative abundance of high-rank bacterial taxa”

Responses 27: As per your suggestion, we have revised “Bacterial abundance stacking” to “Relative abundance of high-rank bacterial taxa” in the manuscript, Line 271-272.

Comments 28: Line 241: Figure 2a: What does the blue-pink scale on the right show? Add a signature.

Responses 28: The blue-pink scale on the right indicates “Time”. We have added the information into Figure 1(a) and incorporated the corresponding description in the revised manuscript, Line 287-288.

Comments 29: Line 261: Figure 3: You could write the abbreviated name of the major taxon (phylum or class) next to the genus names to make it clearer. Or choose some other way to indicate which taxa these genera belong to.

Responses 29: Thank you for your opinion. We have made modifications in Figure 3.

Comments 30: Line 276: “mantel” should be changed to “Mantel”

Responses 30: The word “mantel” has been changed to “Mantel”, as can been seen in the revised manuscript in Line 320.

Comments 31: Line 289-290: “A neutral model-based analysis at the family level was conducted in the upper, middle, and lower sections” – What are upper, middle, and lower sections? These are shown in Figure S1 but it is not clear what they mean.

Responses 31: The upper, middle, and lower sections refer to the upper, middle, and lower segments of the dark blue curve in Figure 5 of the neutral model. Specially, these sections represent the OTUs occurring more frequently than predicted, OTUs considered neutrally distributed, and OTUs occurring less frequently than predicted, respectively. Furthermore, the bacterial communities in each of these three sections were filtered out to create family-level relative abundance stacking plots, as displayed in Figure S2. Relevant information has been added in the revised manuscript, Line 333-335, and title of Figure S2.

Comments 32: Line 290: Figure S1: In supplementary figure, this is what the indication of the authors of your article looks like: “Firstname Lastname 1, Firstname Lastname 2 and Firstname Lastname 2,*”. Tidy up the supplementary figure.

Responses 32: Thanks for pointing out the issue. We have modified it in supplementary figure.

Comments 33: “Figure S1. Taxonomic distribution of OTUs (at the family level) for the four groups.” – Write a clearer description of what this picture shows. What does Above, Neutral, Below mean?

Responses 33: Relevant descriptions have been incorporated into the revised title of Figure S2.

4. Discussion

Comments 34: Line 316-318: “Our findings revealed that N. oculata exerted a more pronounced influence on nutrient levels than T. weissflogii and the combination of the two species.” – In your article there is no data on changes in the number and biomass of microalgae during the experiment. It is not known whether the T. weissflogii culture continued to grow, or whether it died after inoculation, so there was a low impact on bacterial communities.

Responses 34: The changes in microalgal biomass have been incorporation into the supplementary figures as Figure S1.

Comments 35: Line 325-326: Microalgae exert a strong influence on the diversity and composition of bacterial communities, as indicated in previous studies [23]. 23. Datta, M.S.; Sliwerska, E.; Gore, J.; Polz, M.F.; Cordero, O.X. Microbial Interactions Lead to Rapid Micro-Scale Successions on Model Marine Particles. – this article is about bacterial colonization of particle organic matter. I checked through a search and did not find the word “microalgae” in this article.

Responses 35: We have revised this in the manuscript to reference [32] (32.   Kimbrel, J.A. Host Selection and Stochastic Effects Influence Bacterial Community Assembly on the Microalgal Phycosphere. Algal Research 2019, 40, 101489, doi:10.1016/j.algal.2019.101489.), as indicated in Line 371 and Line 595-596.

Comments 36: Line 332-336: “This finding aligns with previous observations where microalgae, specifically N. oculata, influenced the intestinal bacterial community of European seabass. N. oculata increased the representation of Bacillus, known for their probiotic potential, and reduced the abundance of potentially pathogenic bacteria such as Acinetobacter [24]. 24. Ferreira et al. Gracilaria Gracilis and Nannochloropsis Oceanica, Singly or in Combination, in Diets Alter the Intestinal Microbiota of European Seabass (Dicentrarchus Labrax). – this article is about Nannochloropsis oceanica, not N. oculata, as you write.

Responses 36: The reference has been updated to reference [27] for the impact of microalgae on bacterial communities. (27.        Ding, Y.; Chen, N.; Ke, J.; Qi, Z.; Chen, W.; Sun, S.; Zheng, Z.; Xu, J.; Yang, W. Response of the Rearing Water Bacterial Community to the Beneficial Microalga Nannochloropsis Oculata Cocultured with Pacific White Shrimp (Litopenaeus Vannamei). Aquaculture 2021, 542, 736895, doi:10.1016/j.aquaculture.2021.736895.), as indicated in Line 379 and Line 582-584.

Comments 37: Line 337-338: Chlorophyta influence bacterial communities through metabolites [4]. 4. Amin et al. Interaction and Signalling between a Cosmopolitan Phytoplankton and Associated Bacteria. – this article is about interaction between diatom and bacteria. This article does not examine the influence of Chlorophyta on bacterial communities.

Responses 37: The reference has been updated to reference [33] (33.     Alsufyani, T.; Weiss, A.; Wichard, T. Time Course Exo-Metabolomic Profiling in the Green Marine Macroalga Ulva (Chlorophyta) for Identification of Growth Phase-Dependent Biomarkers. Marine Drugs 2017, 15, 14, doi:10.3390/md15010014.), as indicated in Line 381 and Line 597-599.

Comments 38: Line 342-343: In microalgae-inoculated waters, T. weissflogii was symbiotic with Marivita [26].

What is the reason for the emphasis on Marivita? It was not shown in your work, but in the article “26. Sun et al. Diatom Red Tide Significantly Drives the Changes of Microbiome in Mariculture Ecosystem”, in addition to this genus, there were several more genera of symbiotic bacteria.

Responses 38: The inaccurate expression has been revised to “In microalgae-inoculated waters, T. weissflogii could form a symbiotic relationship with specific bacteria”, as indicated in the revised manuscript, Line 385-386.

Comments 39: Line 347-350: “When the nitrogen source is ammonium, the oxygen generated by microalgae has a positive influence on nitrifying bacteria [27]. Conversely, when nitrate serves as the nitrogen source, oxygen can have an inhibitory effect on denitrifying bacteria [27].”

In your work, have nitrifying and denitrifying bacteria been identified in bacterial communities? If not, then why discuss them.

Responses 39: Both nitrifying and denitrifying bacteria have been identified as remarkable bacteria, as illustrated in Figure S3. The manuscript has been revised to provide an explanation for the inclusion of these two types of bacteria, as detailed in Line 390-393.

Comments 40: Line 354-357: In a lake ecosystem, the FL bacterial community played a critical role in methane utilization, while the PA community seemed to have contributed more to biogeochemical cycling [31]. 31. Shen, et al. Similar Assembly Mechanisms but Distinct Co-Occurrence Patterns of Free-Living vs. Particle-Attached Bacterial Communities across Different Habitats and Seasons in Shallow, Eutrophic Lake Taihu.

In this work, it was in Lake Taihu that the FL bacterial community played a critical role in methane utilization, but in other lakes FL bacteria may play other roles, so it probably should not be generalized to “a lake ecosystem.” Also, in Shen, et al. it is written that “the PA bacteria contributed more to biogeochemical cycling of carbon.”, but you write “the PA community seemed to have contributed more to biogeochemical cycling.” It is necessary to cite literature sources more accurately and correctly.

Responses 40: Thank you very much for your careful review. We have changed the original sentence "In a lake ecosystem, the FL bacterial community played a critical role in methane utilization, while the PA community seemed to have contributed more to biogeochemical cycles". have contributed more to biogeochemical cycling [31]" was replaced with "Some previous studies have demonstrated that FL and PA bacteria are significantly different from each other [16,43]."(16.   Crespo, B.G.; Pommier, T.; Fernández-Gómez, B.; Pedrós-Alió, C. Taxonomic Composition of the Particle-Attached and Free-Living Bacterial Assemblages in the Northwest Mediterranean Sea Analyzed by Pyrosequencing of the 16S rRNA. MicrobiologyOpen 2013, 2, 541–552, doi:10.1002/mbo3.92. 43. Bachmann, J.; Heimbach, T.; Hassenrück, C.; Kopprio, G.A.; Iversen, M.H.; Grossart, H.P.; Gärdes, A. Environmental Drivers of Free-Living vs. Particle-Attached Bacterial Community Composition in the Mauritania Upwelling System. Frontiers in Microbiology 2018, 9, 2836, doi:10.3389/fmicb.2018.02836.), as can be seen in the revised manuscript, Line 400-401.

Comments 41: Line 361-362: “PA bacteria have richer bacterial communities than FL bacteria” What does “richer” mean?

Responses 41: The term "richer" refers to a higher richness of species in the PA bacteria compared to the FL bacteria. The expression of this sentence has been updated to “PA bacteria exhibit a higher richness in bacterial communities compared to FL bacteria”, as can be seen in the revised manuscript, Line 406-407.

Comments 42: Line 374-375: “Deterministic processes are considered the main drivers shaping microecological communities in aquatic environments [35].” What is “microecological communities”?

35. Cruaud, et al. Annual Protist Community Dynamics in a Freshwater Ecosystem Undergoing Contrasted Climatic Conditions: The Saint-Charles River (Canada). – This article is about protist (microeukaryotic) communities in river. The phrases “Deterministic processes” and “microecological communities” are missing from this article.

Responses 42: The term “microecological community” has been revised to “microbial community”. Additionally, to address the ambiguity in this sentence, we have made corrections in the revised manuscript and cited the relevant references [48-52] (48. Chen, W.; Ren, K.; Isabwe, A.; Chen, H.; Liu, M.; Yang, J. Stochastic Processes Shape Microeukaryotic Community Assembly in a Subtropical River across Wet and Dry Seasons. Microbiome 2019, 7, 138, doi:10.1186/s40168-019-0749-8. 49. Liu, L.; Yang, J.; Yu, Z.; Wilkinson, D.M. The Biogeography of Abundant and Rare Bacterioplankton in the Lakes and Reservoirs of China. ISME J 2015, 9, 2068–2077, doi:10.1038/ismej.2015.29. 50. Zhao, D.; Xu, H.; Zeng, J.; Cao, X.; Huang, R.; Shen, F.; Yu, Z. Community Composition and Assembly Processes of the Free-Living and Particle-Attached Bacteria in Taihu Lake. FEMS Microbiology Ecology 2017, 93, doi:10.1093/femsec/fix062. 51. Yan, Q.; Stegen, J.C.; Yu, Y.; Deng, Y.; Li, X.; Wu, S.; Dai, L.; Zhang, X.; Li, J.; Wang, C.; et al. Nearly a Decade-Long Repeatable Seasonal Diversity Patterns of Bacterioplankton Communities in the Eutrophic Lake Donghu (Wuhan, China). Molecular Ecology 2017, 26, 3839–3850, doi:10.1111/mec.14151. 52. Wang, J.; Shen, J.; Wu, Y.; Tu, C.; Soininen, J.; Stegen, J.C.; He, J.; Liu, X.; Zhang, L.; Zhang, E. Phylogenetic Beta Diversity in Bacterial Assemblages across Ecosystems: Deterministic versus Stochastic Processes. ISME J 2013, 7, 1310–1321, doi:10.1038/ismej.2013.30.).

Details can be found in the revised manuscript, Line 419-422 and Line 639-652.

Comments 43: Line 375-378: “It has been shown that stochastic processes dominate pathogenic community assembly, while deterministic ecological niche processes are the dominant mechanisms controlling pathogenic community assembly [36].” – This sentence was taken out of context and therefore lost its meaning. 36. Zhang, et al. Differences in Pathogenic Community Assembly Processes and Their Interactions with Bacterial Communities in River and Lake Ecosystems – The original sentence looks like this: “Stochastic processes dominate pathogenic community assembly in riverine habitats, while deterministic ecological niche processes are the dominant mechanisms controlling pathogenic community assembly in the lake habitat.”

Responses 43: Thank you very much for your comment. We have revised this sentence in the manuscript due to its problematic nature and cited the relevant references [48-52] (48. Chen, W.; Ren, K.; Isabwe, A.; Chen, H.; Liu, M.; Yang, J. Stochastic Processes Shape Microeukaryotic Community Assembly in a Subtropical River across Wet and Dry Seasons. Microbiome 2019, 7, 138, doi:10.1186/s40168-019-0749-8. 49. Liu, L.; Yang, J.; Yu, Z.; Wilkinson, D.M. The Biogeography of Abundant and Rare Bacterioplankton in the Lakes and Reservoirs of China. ISME J 2015, 9, 2068–2077, doi:10.1038/ismej.2015.29. 50. Zhao, D.; Xu, H.; Zeng, J.; Cao, X.; Huang, R.; Shen, F.; Yu, Z. Community Composition and Assembly Processes of the Free-Living and Particle-Attached Bacteria in Taihu Lake. FEMS Microbiology Ecology 2017, 93, doi:10.1093/femsec/fix062. 51. Yan, Q.; Stegen, J.C.; Yu, Y.; Deng, Y.; Li, X.; Wu, S.; Dai, L.; Zhang, X.; Li, J.; Wang, C.; et al. Nearly a Decade-Long Repeatable Seasonal Diversity Patterns of Bacterioplankton Communities in the Eutrophic Lake Donghu (Wuhan, China). Molecular Ecology 2017, 26, 3839–3850, doi:10.1111/mec.14151. 52. Wang, J.; Shen, J.; Wu, Y.; Tu, C.; Soininen, J.; Stegen, J.C.; He, J.; Liu, X.; Zhang, L.; Zhang, E. Phylogenetic Beta Diversity in Bacterial Assemblages across Ecosystems: Deterministic versus Stochastic Processes. ISME J 2013, 7, 1310–1321, doi:10.1038/ismej.2013.30.), Line 419-422 and Line 639-652.

Comments 44: Line 403-404: “Our results show that T. weissflogii had a positive effect on more bacteria than Microcystis aeruginosa, such as Saprospiraceae.” What does this mean? Your work does not evaluate the effect of Microcystis aeruginosa on bacteria, including Saprospiraceae.

Responses 44: The error “Microcystis aeruginosa” has been corrected to “N. oculate”, as indicated in Line 444.

Comments 45 Line 409-411: “Coincidentally, it has been demonstrated that the diatom phylum dominates changes in bacterial community structure [43].” again incorrect quotation. 43. Klamt, et al. An Extreme Drought Event Homogenises the Diatom Composition of Two Shallow Lakes in Southwest China. – this article is only about diatoms. It does not study the effect of diatoms on bacteria. This article doesn't even contain the phrase “bacterial community.”

Responses 45: The original reference has been updated to reference [55] (55.        Sison-Mangus, M.P.; Jiang, S.; Kudela, R.M.; Mehic, S. Phytoplankton-Associated Bacterial Community Composition and Succession during Toxic Diatom Bloom and Non-Bloom Events. Frontiers in Microbiology 2016, 7, 1433, doi:3389.2016/fmicb.01433.), Line 450 and Line 659-661.

Reviewer 2 Report

Comments and Suggestions for Authors

This study highlights the impact of microalgae, specifically Nannochloropsis oculata and Thalassiosira weissflogii, on bacterial communities in shrimp-farming waters. The findings suggest that adding microalgae, especially T. weissflogii, influences the structure and stability of particle-attached bacterial communities, revealing potential benefits for regulating microbial dynamics in shrimp aquaculture ecosystems. The emphasis on environmental factors as primary influencers and insights into community changes contribute to understanding and promoting sustainable development in shrimp aquaculture while considering ecosystem conservation. The article must be improved before being considered for publication:

1.     Introduction:

a.     The introduction mentions the adverse environmental consequences of intensive shrimp rearing but does not tie it back to the subsequent discussion on bacterial communities and microalgae. A more direct connection between the initial environmental concerns and the subsequent focus on bacterial dynamics would enhance coherence.

b.     Some statements, such as "Aquaculture has experienced remarkable growth in recent years," are out of context since the reference for this statement dates from 2010, which is not recent.

2.     Methodology

a.     The initial biomass of microalgae is mentioned to be approximately 25 mg/L, but it would be beneficial to elaborate on how this specific concentration was determined and whether it reflects natural conditions in shrimp aquaculture ponds.

b.     The description of the water sample collection is detailed. However, it would be helpful to know how the samples were handled to prevent contamination during the collection process.

c.     While the DNA extraction method is mentioned, the rationale for using the MinkaGene Water DNA kit could be explained. Additionally, details about how potential inhibitors of downstream processes were addressed during DNA extraction could be beneficial.

d.     Further details on how the AVD index was calculated and interpreted would enhance the understanding of community stability assessment.

3.     Discussion and Conclusion

a.     The discussion states that adding T. weissflogii resulted in a more stable bacterial community but later mentions that both T. weissflogii and N. oculata led to community succession and instability. This apparent contradiction needs clarification.

b.      The study generalizes the positive impact of adding specific microalgae on the stability and health of the shrimp aquaculture community. It's essential to acknowledge the conditions under which these results were obtained and consider potential variations in different environments. While the study recognizes its limitations, it would be beneficial to specify the nature of these limitations more explicitly. For example, clarifying the aspects of bacterial community change that the study does not fully explain can guide future research directions.

c.     The discussion links the stability of bacterial communities to the addition of microalgae, suggesting an inhibitory or facilitative effect through metabolites. However, the specific mechanisms and metabolites responsible for these effects are unclear and warrant further investigation.

Comments on the Quality of English Language

Moderate editing of English language required

Author Response

We are very grateful for your affirmation of our work. We greatly appreciate your comments and the thorough revision that greatly contributed to the improvement of our manuscript. Please find the detailed responses below. Red color indicated the corresponding revisions in the re-submitted files.

1. Introduction:

Comments 1: The introduction mentions the adverse environmental consequences of intensive shrimp rearing but does not tie it back to the subsequent discussion on bacterial communities and microalgae. A more direct connection between the initial environmental concerns and the subsequent focus on bacterial dynamics would enhance coherence.

Responses 1: Relevant information has been added, and details can be seen in the revised manuscript, Line 59-61.

Comments 2: Some statements, such as "Aquaculture has experienced remarkable growth in recent years," are out of context since the reference for this statement dates from 2010, which is not recent.

Responses 2: The references have been updated with recent literature [1] (1. Edwards, P.; Zhang, W.; Belton, B.; Little, D.C. Misunderstandings, Myths and Mantras in Aquaculture: Its Contribution to World Food Supplies Has Been Systematically over Reported. Marine Policy 2019, 106, 103547, doi:10.1016/j.marpol.2019.103547.), as can be seen in Line 53 and Line 506-508.

2. Methodology

Comments 3: The initial biomass of microalgae is mentioned to be approximately 25 mg/L, but it would be beneficial to elaborate on how this specific concentration was determined and whether it reflects natural conditions in shrimp aquaculture ponds.

Responses 3: The initial biomass of these two microalgae was determined based on the microalgal biomass of shrimp pond at Xiangshan Lanshang Marine Technology Co., Ltd., from where the shrimp rearing water and shrimp samples were obtained. Relevant information has been added in the revised manuscript, Line 137.

Comments 4: The description of the water sample collection is detailed. However, it would be helpful to know how the samples were handled to prevent contamination during the collection process.

Responses 4: All instruments were autoclaved before use during sample collection. Filter membranes were rapidly frozen in liquid nitrogen and stored at -80 °C after collection. Water samples are stored in sterile centrifuge tubes in a refrigerator at -4 °C.

Comments 5: While the DNA extraction method is mentioned, the rationale for using the MinkaGene Water DNA kit could be explained. Additionally, details about how potential inhibitors of downstream processes were addressed during DNA extraction could be beneficial.

Responses 5: During processing, a flocculant is added to adsorb impurities, and then centrifugation precipitates the flocculant and impurities to increase the purity of the nucleic acids. At the same time, the inhibitors are also adsorbed together. Relevant information has been added in the revised manuscript, Line 174-177.

Comments 6: Further details on how the AVD index was calculated and interpreted would enhance the understanding of community stability assessment.

Responses 6: Community stability was assessed using the average variation degree (AVD) index, calculated as the deviation degree from the mean of normally distributed OTU relative abundances among different groups. Lower AVD value indicates higher microbiome stability. The variation degree for each OTU was calculated using the following equation (Eq: ), in which ai is the variation degree for an OTU, xi is the rarefied abundance of the OTU in one sample, Ì…xi is the average rarefied abundance of the OTU in one sample group, and δi is the standard deviation of the rarefied abundances of the OTU in one sample group. The AVD values were calculated using the following equation (Eq: ), in which k is the number of samples in one sample group, n is the number of OTUs in each sample group. Specialized metabolic functions of keystone taxa sustain soil microbiome stability. The relevant reference is [31] (31.      Xun, W.; Liu, Y.; Li, W.; Ren, Y.; Xiong, W.; Xu, Z.; Zhang, N.; Miao, Y.; Shen, Q.; Zhang, R. Specialized Metabolic Functions of Keystone Taxa Sustain Soil Microbiome Stability. Microbiome 2021, 9, 35, doi:10.1186/s40168-020-00985-9.), Line 229-236 and Line 592-594.

3. Discussion and Conclusion

Comments 7: The discussion states that adding T. weissflogii resulted in a more stable bacterial community but later mentions that both T. weissflogii and N. oculata led to community succession and instability. This apparent contradiction needs clarification.

Responses 7: Thank you very much for your thorough review. What we intended to convey is that it can be seen in Figure 6 indicated that the bacterial community of group M (T. weissflogii and N. oculata) is not as stable as that of group T (T. weissflogii). T. weissflogii will stabilize the bacterial community, but when T. weissflogii and N. oculata are mixed and added, it will cause the bacterial community to undergo succession, making the community relatively instable. To avoid any potential misunderstandings, we have made specific changes, Line 447.

Comments 8: The study generalizes the positive impact of adding specific microalgae on the stability and health of the shrimp aquaculture community. It's essential to acknowledge the conditions under which these results were obtained and consider potential variations in different environments. While the study recognizes its limitations, it would be beneficial to specify the nature of these limitations more explicitly. For example, clarifying the aspects of bacterial community change that the study does not fully explain can guide future research directions.

Responses 8: We have incorporated this limitation into the Conclusion section, as can be seen in the revised manuscript, Line 467-474.

Comments 9: The discussion links the stability of bacterial communities to the addition of microalgae, suggesting an inhibitory or facilitative effect through metabolites. However, the specific mechanisms and metabolites responsible for these effects are unclear and warrant further investigation.

Responses 9: Thank you very much for your careful review. We have pointed out that the specific mechanisms and metabolites responsible for these effects are unclear and warrant further investigation, Line 467-474.

Reviewer 3 Report

Comments and Suggestions for Authors The article by Lian et al. is devoted to optimizing shrimp farming conditions by creating algal-bacterial communities in the culture environment. This methodological approach is based on scientific literature data on the difference in taxonomic structure of free-living (FL) bacterial communities and bacteria attached to particles (PA), in this case the cells of two species of diatoms. In the Introduction chapter, it would be a good idea to add references from recent years on the taxonomic difference between FL and PA bacteria in order to emphasize the breadth and fundamental nature of this topic. To achieve the goal, the authors applied modern research methods and statistical data processing. The methods are well described. The results are well illustrated. The conclusions are supported by the results obtained.

The article may be accepted for publication in Biology with minor edits.

I think that the meaning of the work is broader than what the authors write in the last sentence of the Abstract: “This study is important for understanding the microbial communities in shrimp aquaculture ecosystems and may contribute to the sustainable development of shrimp aquaculture and ecosystem conservation.” In addition to its undoubted practical significance, the work is of fundamental importance, as it shows the complexity of interactions between aquatic organisms and shows different consequences for the introduction of additional participants in zoo-phyto-bacterioplankton interactions.

Figure 1b is difficult to understand. It is necessary to expand the caption to the figure; it is not indicated in what units time is measured. And the layout of Figure 1b itself is not good; it is difficult to compare one with the other. I would recommend cutting it into 8 parts, C, N, T, M placed vertically, one below the other, the PA column on the left, FL on the right.

I would not have written the last phrase in Conclusion “However, this study has limitations and does not fully explain the process of bacterial community change, and further research is needed.” Obviously, further research is needed in any work; in addition, the purpose of this article was not to explain the process of bacterial community change.

Some attention is required to the design of references in the list of references in terms of writing the titles of the cited articles and the names of journals.

Author Response

We are very grateful for your affirmation of our work. We greatly appreciate your comments and the thorough revision that greatly contributed to the improvement of our manuscript. Please find the detailed responses below. Red color indicated the corresponding revisions in the re-submitted files.

Comments 1: I think that the meaning of the work is broader than what the authors write in the last sentence of the Abstract: “This study is important for understanding the microbial communities in shrimp aquaculture ecosystems and may contribute to the sustainable development of shrimp aquaculture and ecosystem conservation.” In addition to its undoubted practical significance, the work is of fundamental importance, as it shows the complexity of interactions between aquatic organisms and shows different consequences for the introduction of additional participants in zoo-phyto-bacterioplankton interactions.

Responses 1: The abstract was kept concise as it was limited to 200 words. However, in response to your suggestion, we have added at the end of the abstract “This study is important for understanding the microbial community assembly and interpreting complex interactions among zoo-phyto-bacterioplankton in shrimp aquaculture ecosystems. Additionally, these findings may contribute to the sustainable development of shrimp aquaculture and ecosystem conservation.”

Comments 2: Figure 1b is difficult to understand. It is necessary to expand the caption to the figure; it is not indicated in what units time is measured. And the layout of Figure 1b itself is not good; it is difficult to compare one with the other. I would recommend cutting it into 8 parts, C, N, T, M placed vertically, one below the other, the PA column on the left, FL on the right.

Responses 2: The “time” in the figure refers to the number of days the experiment was conducted. The relevant information has been added to the revised manuscript, Line 272. Regarding the layout of the figures, we appreciate your suggestion. To emphasize the differences in bacterial community among different groups, after careful consideration, we have decided not to create eight separate figures for the PA and FL bacterial communities of the C, N, T, M group. However, we have made further modifications to Fig. 1(b) to present FL and PA bacteria in different forms for a more intuitive interpretation.

Comments 3: I would not have written the last phrase in Conclusion “However, this study has limitations and does not fully explain the process of bacterial community change, and further research is needed.” Obviously, further research is needed in any work; in addition, the purpose of this article was not to explain the process of bacterial community change.

Responses 3: Based on your feedback, we have revised our Conclusions to better reflect the goals and results of our study. The updated Conclusions emphasize the impacts on shrimp aquaculture and ecosystem protection and further mention the limitations of the study. We believe that this revision is more in line with the core contributions of our work, Lines 467-474.

Comments 4: Some attention is required to the design of references in the list of references in terms of writing the titles of the cited articles and the names of journals.

Responses 4: The format of references has been thoroughly reviewed and adjusted in accordance with your suggestions and the journal's requirements.

Reviewer 4 Report

Comments and Suggestions for Authors

It is very well-done research and nicely made MS. Congrats.

The main claim of this paper is that the microalgae species tested in this study as well as the nutrient salts play an important role in the assembly and response mechanisms of microbial communities in shrimp aquaculture ecosystems are significant and well-founded.  

The paper stands out from others in the field for its extensive statistical evaluation of the results, although this arsenal of multivariate methods may seem a bit excessive. 

The claims are novel and convincing, although a recent paper by a very similar team (The assembly process of free-living and particle-attached bacterial communities in shrimp-rearing waters: overwhelming influence of nutrient factors relative to microalgal inoculation. Yikai Shi 1, Xuruo Wang 1, Huifeng Cai 2, Jiangdong Ke 1, Jinyong Zhu 1, Kaihong Lu 1, Zhongming Zheng 1 and 5 Wen Yang., Animals, 2023) deals with the same topic. It is puzzling why this article is not cited here because, in my opinion, the results of this work would further strengthen the paper. 

Some field studies would be needed to improve the acceptability of the claims also from the point of view of practical shrimp aquaculturists. This kind of time and resource-consuming study may be difficult to carry out. 

Author Response

We are very grateful for your affirmation of our work. We greatly appreciate your comments and the thorough revision that greatly contributed to the improvement of our manuscript. Please find the detailed responses below. Red color indicated the corresponding revisions in the re-submitted files.

Comments 1: It is very well-done research and nicely made MS. Congrats.

The main claim of this paper is that the microalgae species tested in this study as well as the nutrient salts play an important role in the assembly and response mechanisms of microbial communities in shrimp aquaculture ecosystems are significant and well-founded. 

The paper stands out from others in the field for its extensive statistical evaluation of the results, although this arsenal of multivariate methods may seem a bit excessive.

The claims are novel and convincing, although a recent paper by a very similar team (The assembly process of free-living and particle-attached bacterial communities in shrimp-rearing waters: overwhelming influence of nutrient factors relative to microalgal inoculation. Yikai Shi 1, Xuruo Wang 1, Huifeng Cai 2, Jiangdong Ke 1, Jinyong Zhu 1, Kaihong Lu 1, Zhongming Zheng 1 and 5 Wen Yang., Animals, 2023) deals with the same topic. It is puzzling why this article is not cited here because, in my opinion, the results of this work would further strengthen the paper.

Responses 1: Thank you very much for your approval of our research work. We agree with your suggestions. This manuscript involves an extended validation of the experiment conducted by Shi et al. (2023). We have incorporated this preliminary research the introduction section, as can be seen in the revised manuscript, Line 98-99.

Comments 2: Some field studies would be needed to improve the acceptability of the claims also from the point of view of practical shrimp aquaculturists. This kind of time and resource-consuming study may be difficult to carry out.

Responses 2: Your suggestion is highly valuable. Despite our experiment, which operated in a pilot-scale facility and to some extent replicated aquaculture practices, there are limitations related to the absence of real aquaculture pones and potential drawbacks in simulating aquaculture practices. We have addressed these limitations in the Conclusion section to caution other readers, as can be seen in the revised manuscript, Line 467-474. Additionally, we intend to conduct this experiment again in actual aquaculture ponds to further validate our findings.

Round 2

Reviewer 1 Report

Comments and Suggestions for Authors

 The authors have done a good, precise job of improving the manuscript.

But I have some minor comments.

Line 129-134. In Responses 16: “Considering the difference in biological volume between N. oculata and T. weissflogii, we chose the microalgal biomass as the indicator for experiment design and results presentation instead of density.” I think it is important information and should be included in the article.

 Line 139-141: Was the shrimp culture constantly/periodically stirred or not?

 Line 193-194: “minsize = 4”

In Responses 21: “The “minsize” is another parameter used with unoise3 or unoise2, and its purpose is to set the minimum size (length) of the denoised sequences to be retained. Sequences shorter than this minimum size are discarded. Both parameters aid in filtering out very short sequences that may result from sequencing errors or other artifacts.”

 minsize (minimum size) is not length. From Usearch manual for unoise3 command: “The -minsize option specifies the minimum abundance. Default is 8. Input sequences with lower abundances are discarded. For higher sensivity, reducing minsize to 4 is reasonable, especially if samples are denoised indivudually rather pooling all samples together, as I would usually recommend.”

If you use Usearch in your subsequent works, it is better to add the “minLen” parameter – the minimum length of sequences. Sequences shorter than the specified minLen values ​​are removed. If you do not specify the value of this parameter, then very short sequences may be included in the analysis, which are noise for this data. With minLen, analysis becomes higher quality and more rigorous.

By the way, the description for the unoise3 command states: “The -unoise_alpha option specifies the alpha parameter. Default is 2.0.” If you use unoise_alpha = 2, which is the default, then you probably don't need to specify it in the command.

 Line 196-199. “The low-quality reads, singletons and chimeric sequences were decarded during sequence analysis process prior to clustered ZOTUs. Assignments of representative sequences to each ZOTU were performed using the RDP classifier, utilizing the SILVA bacterial data-base (version v138) with a 99% similarity threshold.”

Nowhere is it stated that zOTUs of chloroplasts were removed.

 Table S1 Could you add a column with the number of reads (sequences)? Or add ranges of the number of reads to the text of the article. Without this information, it is not clear how large your data is in order to determine the appropriate richness in it.

 Line 330: “OTU” OTU or ZOTU?

 Line 333-335: “A neutral model-based analysis at the family level was conducted in the upper (OTUs occurring more frequently than predicted), middle (OTUs considered neutrally distributed), and lower (OTUs occurring less frequently than predicted) sections”

Have you analyzed OTUs or ZOTUs?

Figure S2. The bar chart (B) shows the taxonomic distribution (family level) of OTUs in three parts (above prediction, below prediction, and neutral distribution) predicted by the four sets of neutral models.

 Line 362: “species ().”Did you forget to remove the parentheses or should there be something inserted there?

 Line 483: “OTU” 

Author Response

We are very grateful for your affirmation of our work. We greatly appreciate your comments and the thorough revision that greatly contributed to the improvement of our manuscript. Please find the detailed responses below. Red color indicated the corresponding revisions in the resubmitted files.

Comments 1:

The authors have done a good, precise job of improving the manuscript.

But I have some minor comments.

Line 129-134. In Responses 16: “Considering the difference in biological volume between N. oculata and T. weissflogii, we chose the microalgal biomass as the indicator for experiment design and results presentation instead of density.” I think it is important information and should be included in the article.

Responses 1: Thank you very much for your suggestion. We have added this sentence “Given the substantial difference in biological volume between N. oculata and T. weissflogii, microalgal biomass was opted as the indicator for experiment design and results presentation instead of abundance.” in the revised manuscript, Line 129-131.

Comments 2: Line 139-141: Was the shrimp culture constantly/periodically stirred or not?

Responses 2: Line 144 of the text mentions that electric aerators were used to aerate all tanks. The shrimp culture was consistently aerated with electric aerators, ensuring continuous water flow and, consequently, constant stirring of the shrimp culture.

Comments 3: Line 193-194: “minsize = 4”

In Responses 21: “The “minsize” is another parameter used with unoise3 or uniose2, and its purpose is to set the minimum size (length) of the denoised sequences to be retained. Sequences shorter than this minimum size are discarded. Both parameters aid in filtering out very short sequences that may result from sequencing errors or other artifacts.”

minsize (minimum size) is not length. From Usearch manual for unoise3 command: “The -minsize option specifies the minimum abundance. Default is 8. Input sequences with lower abundances are discarded. For higher sensivity, reducing minsize to 4 is reasonable, especially if samples are denoised indivudually rather pooling all samples together, as I would usually recommend.”

If you use Usearch in your subsequent works, it is better to add the “minLen” parameter – the minimum length of sequences. Sequences shorter than the specified minLen values are removed. If you do not specify the value of this parameter, then very short sequences may be included in the analysis, which are noise for this data. With minLen, analysis becomes higher quality and more rigorous.

By the way, the description for the unoise3 command states: “The -unoise_alpha option specifies the alpha parameter. Default is 2.0.” If you use unoise_alpha = 2, which is the default, then you probably don't need to specify it in the command.

Responses 3: Your expert advice on data analysis is greatly appreciated, and we have found it immensely valuable. In our future analyses, we will incorporate more rigorous procedures. We also acknowledge the importance of avoiding unnecessary parameter specifications when default values can be used for better clarity. Thank you for your guidance.

Comments 4: Line 196-199. “The low-quality reads, singletons and chimeric sequences were decarded during sequence analysis process prior to clustered ZOTUs. Assignments of representative sequences to each ZOTU were performed using the RDP classifier, utilizing the SILVA bacterial data-base (version v138) with a 99% similarity threshold.”

Nowhere is it stated that zOTUs of chloroplasts were removed.

Responses 4: We have removed ZOTUs of singletons, chimeric, mitochondria, and chloroplasts from the data. To prevent any misunderstandings and confusion, we have included relevant information in the revised manuscript, Line 198.

Comments 5: Table S1 Could you add a column with the number of reads (sequences)? Or add ranges of the number of reads to the text of the article. Without this information, it is not clear how large your data is in order to determine the appropriate richness in it.

Responses 5: According to your comments, we have added a column in Table S1 to display the number of reads, highlighted in red. The values in this column represent the average and standard deviation of the number of reads in each group.

Comments 6: Line 330: “OTU” OTU or ZOTU?

Responses 6: Thank you for your attention. We have corrected the term “OTU” to “ZOTU” in the revised manuscript, Line 333.

Comments 7: Line 333-335: “A neutral model-based analysis at the family level was conducted in the upper (OTUs occurring more frequently than predicted), middle (OTUs considered neutrally distributed), and lower (OTUs occurring less frequently than predicted) sections”

Have you analyzed OTUs or ZOTUs?

Figure S2. The bar chart (B) shows the taxonomic distribution (family level) of OTUs in three parts (above prediction, below prediction, and neutral distribution) predicted by the four sets of neutral models.

Responses 7: We apologize for the oversight. We did analyze ZOTUs, and we have corrected the related issues in the revised manuscript.

Comments 8: Line 362: “species ().”Did you forget to remove the parentheses or should there be something inserted there?

Responses 8: Thank you for bringing this error to our attention. The parentheses have been removed in the revised manuscript, Line 365.

Comments 9: Line 483: “OTU”

Responses 9: We have replaced “OTUs” with “ZOTUs” in the revised manuscript, Line 485.

Reviewer 2 Report

Comments and Suggestions for Authors

The authors complied with all the requested reviews. Great job!

Author Response

We are very grateful for your affirmation of our work. We greatly appreciate your comments and the thorough revision that greatly contributed to the improvement of our manuscript.